# High-Quality Joint Image and Video Tokenization with Causal VAE

**Dawit Mureja Argaw** [1*]   **Xian Liu** [2]    **Qinsheng Zhang** [2]    **Joon Son Chung** [1]
**Ming-Yu Liu** [2]    **Fitsum Reda** [2†]
[1] Korea Advanced Institute of Science and Technology (KAIST)
[2] NVIDIA

## Abstract

Generative modeling has seen significant advancements in image and video synthesis. However, the curse of dimensionality remains a significant obstacle, especially for video generation, given its inherently complex and high-dimensional nature. Many existing works rely on low-dimensional latent spaces from pretrained image autoencoders. However, this approach overlooks temporal redundancy in videos and often leads to temporally incoherent decoding. To address this issue, we propose a video compression network that reduces the dimensionality of visual data both spatially and temporally. Our model, based on a variational autoencoder, employs causal 3D convolution to handle images and videos jointly. The key contributions of our work include a scale-agnostic encoder for preserving video fidelity, a novel spatio-temporal down/upsampling block for robust long-sequence modeling, and a flow regularization loss for accurate motion decoding. Our approach outperforms competitors in video quality and compression rates across various datasets. Experimental analyses also highlight its potential as a robust autoencoder for video generation training. Code and models can be found here.

## 1 Introduction

Recently, generative modeling has taken the world by storm, showcasing remarkable advancements in various domains such as text-to-image (Rombach et al., 2022; Chen et al., 2024; Pernias et al., 2024; Kang et al., 2023; Nguyen et al., 2023), text-to-speech (Le et al., 2023; Wang et al., 2023; Shen et al., 2024) and most recently, text-to-video generation (Blattmann et al., 2023c; Brooks et al., 2024; Blattmann et al., 2023a). Following from the seminal work (Rombach et al., 2022), many generative models (Blattmann et al., 2023c; Brooks et al., 2024; Blattmann et al., 2023a; Yu et al., 2023c; Hu et al., 2023a) now rely on a low-dimensional latent space to achieve high-resolution synthesis. This strategic choice stems from the need to combat the curse of dimensionality in high-resolution synthesis. In video generation, the importance of acquiring a robust low-dimensional latent representation is further emphasized by the complex and high-dimensional nature of video data.

Most works on video compression used in a generation context typically fall into two main categories based on the nature of their latent space: discrete or continuous. While discrete latent spaces have been an active area of investigation, with recent works like (Yu et al., 2024; Gupta et al., 2023) enhancing compression and reconstruction quality, there has been comparatively less exploration of continuous latent spaces. For example, recent video diffusion models (Blattmann et al., 2023c;a; Hu et al., 2023a) still heavily rely on pretrained image autoencoders with added 3D convolutions for temporal alignment, resulting in lingering temporal artifacts such as flickering in decoded videos. This reliance on image autoencoders also means that temporal compression is often overlooked. Consequently, the well-documented temporal redundancy of videos (Lu et al., 2019; Lin et al., 2020; Hu et al., 2023b) is not fully exploited in continuous video autoencoders. This, in turn, has significantly limited the ability of state-of-the-art models (Blattmann et al., 2023c;a; Hu et al., 2023a) to effectively encode and generate longer-duration videos, until the recently introduced Sora (Brooks et al., 2024) model. In light of this, our work introduces a video compression network in continuous

---

*Work done during a research internship at NVIDIA.
†Corresponding author.

time-space that reduces the dimensionality of visual data into a learned latent and maps the generated latent back to pixel space with high fidelity.

Our video compression network is based on a variational autoencoder (VAE) (Kingma & Welling, 2013), where the encoder compresses the input video both spatially and temporally into a latent representation and the decoder reconstructs the input video from this latent representation. We explored both Transformer-based (Vaswani et al., 2017) and 3D convolution-based architectures for a video VAE. Our experiments have shown that a fully Transformer-based video VAE tends to introduce blocking artifacts and struggles to generalize effectively across various temporal and spatial resolutions compared to 3D convolutions, consistent with findings in Yu et al. (2024). Therefore, we opt for 3D convolutions interleaved with self-attention layers to design our video VAE. To seamlessly integrate joint image and video compression within a single model, we use temporally causal 3D convolutions and self-attention layers. In addition to introducing a continuous video VAE for high-quality spatio-temporal compression, our main contributions lie in identifying and addressing three key issues in video VAEs.

**First**, we observe that a standard video VAE with *symmetric* encoder-decoder architecture struggles to maintain the fidelity of a video when it contains small and fast-moving objects, especially at higher compression rates. This is mainly because small, fast-moving objects tend to disappear at the deeper levels of the encoder feature pyramid. Moreover, at these levels, the significantly smaller feature dimension compared to the input dimension makes it difficult to preserve large motion information. To mitigate this challenge, we draw inspiration from FILM (Reda et al., 2022) and propose a weight-shared encoder that learns to aggregate features across different scales of the input video (refer to Fig. 2(a)). The intuition here is that large motion at finer scales (higher resolution) should be equivalent to small motion at coarser scales (lower resolution). Thus, sharing encoder weights and aggregating features from different depths of the feature pyramid allows us to increase the number of pixels available to effectively encode large motion. Our experiments show that incorporating a FILM encoder into a video VAE significantly improved the decoding of large motion.

**Second**, we carefully examine spatio-temporal downsampling and upsampling in video VAEs. A commonly used approach in previous works (Rombach et al., 2022; Yu et al., 2023a) is to employ non-learnable kernels for down/upsampling followed by a convolutional layer *i.e.* average pooling for downsampling and nearest interpolation for upsampling. However, this approach suffers from the potential loss of high-frequency spatio-temporal information, as non-learnable kernels treat all features within the pooling (interpolation) window equally. To address the issue, MAGVIT-v2 (Yu et al., 2024) recently proposed using learnable down/upsampling with convolutional layers. While a video VAE trained with learnable spatio-temporal down/upsampling generally performs well, it tends to overfit to the temporal sequence length it has been trained on, with performance notably dropping when inference is done with different sequence lengths, thus limiting the model's adaptability for arbitrarily long videos. Our work introduces a robust spatio-temporal down/upsampling module that addresses the aforementioned limitations. The module is designed as a dual-path network, effectively leveraging both learnable and non-learnable kernels. We empirically show that the proposed video VAE is *temporally more adaptable* and can encode and decode arbitrarily long videos at varying lengths without significant performance degradation.

**Third**, to ensure that the encoded latent representation faithfully preserves the motion dynamics of the input video, we propose a flow regularization loss for video VAE training. The loss is incorporated by optimizing the mean-squared error between the optical flows of the input video frames and their corresponding optical flows in the decoded video frames. We use a state-of-the-art model (Teed & Deng, 2020) to compute the optical flows. Our experiments reveal that imposing flow regularization loss not only results in the decoding of temporally smoother motion but also helps to achieve a temporal compression rate of up to $16\times$ without a significant loss in decoded video quality.

Building on these findings, we train a causal video VAE at various spatio-temporal compression rates: $4 \times 8 \times 8$ (256), $8 \times 8 \times 8$ (512), $16 \times 8 \times 8$ (1024) and $16 \times 16 \times 16$ (4096). We compare our method with several state-of-the-art approaches (Rombach et al., 2022; Blattmann et al., 2023c; Yan et al., 2021b; Lab & etc., 2024; Zheng et al., 2024; Wang et al., 2024a) across multiple video and image benchmarks (Pont-Tuset et al., 2017; Niklaus & Liu, 2020; Su et al., 2017; Russakovsky et al., 2015), using a comprehensive suite of metrics. Our experimental results demonstrate that the proposed autoencoder consistently outperforms the competing baselines, both qualitatively and quantitatively. To showcase the effectiveness of our video VAE in generative modeling, we integrate

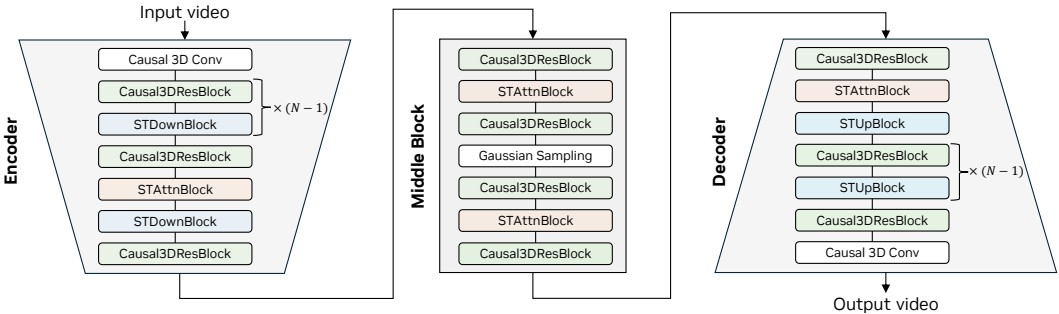

Figure 1: **Architecture Overview**: Our causal video VAE comprises three main components: an encoder, a middle block, and a decoder. The encoder compresses input visual data spatially and temporally, the middle block samples from the learned distribution of encoded features to generate a latent representation, and the decoder maps this latent representation back to pixel space. $N$ denotes the number of down/upsampling steps.

our pretrained autoencoder into open-source video generation frameworks (Wang et al., 2024b; Zheng et al., 2024) and train them for unconditional video generation tasks on the SkyTimelapse (Zhang et al., 2020) and UCF-101 (Soomro, 2012) datasets.

We also perform extensive ablation studies and experimental analyses to further confirm the benefits of the proposed autoencoder. To evaluate the robustness of the video VAE decoder against erroneous latent predictions from generative models, we perturb the encoded latents using various corruption schemes, such as Gaussian noise and quantization, and analyze the model's sensitivity in Sec. B.1.1. Additionally, we assess the video VAE's effectiveness in reliably encoding and decoding videos of varying lengths sampled at different intervals in Sec. B.1.2. Our results strongly affirm the model's potential as a robust autoencoder for training video and image generation models.

## 2 METHOD

We propose a video (image) compression network to reduce the dimensionality of visual data into a learned latent space. Our network is based on a variational autoencoder (VAE) (Kingma & Welling, 2013) architecture consisting of an encoder and decoder. Given a video clip $V$ of size $(1 + T) \times H \times W \times 3$, where $1 + T$ denotes the number of frames in the video, the encoder compresses the input video both temporally and spatially into a latent representation $v$ of dimension $(1 + t) \times h \times w \times c$, with a spatio-temporal compression rate of $\times \frac{T}{t} \cdot \frac{H}{h} \cdot \frac{W}{w}$. The decoder takes the generated latent $v$ as input and maps it back to a video $\hat{V}$ in pixel space. We use temporally causal 3D convolutional and self-attention layers in our network design to seamlessly integrate joint image and video compression within a single model. The overview of our method is depicted in Fig. 1.

### 2.1 ENCODER

The encoder network in the video VAE mainly consists of three components. These are causal 3D residual block (*Causal3DResBlock*), spatio-temporal downsampling block (*STDownBlock*) and spatio-temporal attention block (*STAttnBlock*). As shown in Fig. 1, the encoder is constructed by cascading a stack of these blocks in a top-down fashion.

**Causal 3D Residual Block**   The *Causal3DResBlock* is a residual network composed of two temporally causal 3D convolution layers, along with group normalization (GN) and Swish activation layers (see Fig. 2(b)). For a kernel size $(k_t, k_h, h_w)$, the padding scheme in the temporal axis of a *vanilla* 3D convolutional layer is to add $\lfloor \frac{k_t - 1}{2} \rfloor$ frames before and $\lfloor \frac{k_t}{2} \rfloor$ after the input frames, respectively. In comparison, a *causal* 3D convolution layer pads with $k_t - 1$ frames before the input frames and nothing after, ensuring that the output for each frame solely relies on the preceding frames (Yu et al., 2024; Gupta et al., 2023). As a result, the first frame remains independent of the subsequent frames, enabling our model to compress single images as well. We test three different temporal padding schemes: *zero*, *constant*, and *replication*. Our experiments reveal that employing the replication padding results in a model decoding temporally smoother videos (refer to Sec. B.3).

**Spatio-Temporal Downsampling**   Given a feature volume $x$ of size $b \times (1 + t) \times h \times w \times c$, where $b$ denotes batch size, the *STDownBlock* reduces the spatial and temporal dimensions of $x$ by half, *i.e.* $b \times (1 + \frac{t}{2}) \times \frac{h}{2} \times \frac{w}{2} \times c$. A commonly employed approach in previous works (Rombach

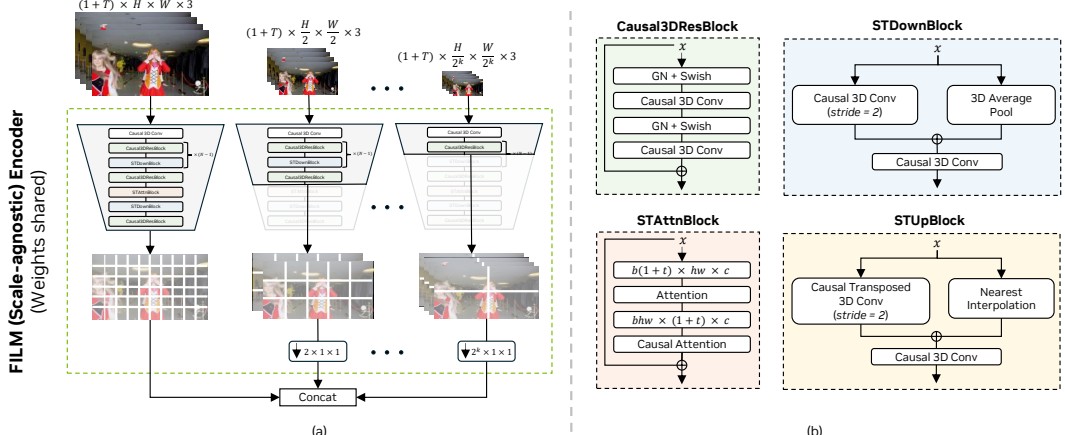

Figure 2: (a) **FILM Encoder**: Our weight-shared encoder aggregates features from multiple scales of the input video, enhancing its ability to handle large motion decoding. (b) **Building Blocks**: Design specifications for the causal 3D residual block (*Causal3DResBlock*), spatio-temporal downsampling block (*STDownBlock*), spatio-temporal upsampling block (*STUpBlock*), and spatio-temporal attention block (*STAttnBlock*).

et al., 2022; Yu et al., 2023a) is to do average pooling followed by a convolution layer. However, such an approach is not optimal, particularly for higher compression rates, because average pooling treats all features within the pooling window equally, potentially resulting in the loss of high-frequency spatial or temporal information. Recently, MAGVIT-v2 (Yu et al., 2024) opted for using strided convolutions instead of average pooling to leverage learned kernels. While a video VAE trained with learnable downsampling kernels generally performs well, it often fails to be *temporally agnostic*, *i.e.* it overfits to the specific sequence lengths it is trained with and performance notably drops when inference is done by sampling the input video at different sequence lengths (see Table 4). This particularly limits the adaptability of the model for long-sequence encoding and the robustness of its latents to noise corruption.

To address these limitations, we introduce a temporally more adaptable downsampling module for a video VAE. We design the *STDownBlock* as a dual-path module, leveraging both learnable and non-learnable kernels, where the input feature is passed through a strided causal 3D convolutional layer and a 3D average pooling layer concurrently, and the resulting outputs are combined via summation as shown in Fig. 2(b). We experimentally observe that the proposed *STDownBlock* effectively addresses the aforementioned limitations and leads to notably better performance. This is intuitive as downsampling via average pooling mitigates the risk of the temporal receptive field overfitting to a specific sequence length while learnable kernels facilitate downsampling by selectively emphasizing or suppressing certain features based on their relevance.

**Spatio-Temporal Attention**   The *STAttnBlock* captures spatial and temporal dependencies within an input video using self-attention layers (Vaswani et al., 2017). Given a feature of size $b \times (1 + t) \times h \times w \times c$, the spatial attention is done by reshaping the feature to $b(1 + t) \times hw \times c$ and passing it to a single self-attention layer. This is followed by causal attention in the temporal dimension by reshaping the input feature to $bhw \times (1 + t) \times c$ as depicted in Fig. 2(b). To manage the increasing computational complexity caused by merging spatial dimensions during spatial attention, we apply the *STAttnBlock* only at the lower resolutions of the feature pyramid in the encoder.

### 2.2   FILM ENCODER

Reducing the dimensionality of video while maintaining its fidelity could be quite challenging particularly when it involves small and fast-moving objects. This is mainly because small objects with large motion disappear at the deeper levels of the encoder feature pyramid. Furthermore, there are significantly fewer pixels at the deeper levels of the pyramid to preserve large motion information. To overcome these challenges, we draw inspiration from FILM (Reda et al., 2022) and design our video VAE with shared encoder weights across different scales as shown in Fig. 2(a). Given a video $V_0$ of size $(1 + T) \times H \times W \times 3$, we first construct an input pyramid $\{V_0, V_1, \ldots, V_k\}$ by successively resizing the video, where $V_k$ has a shape of $(1 + T) \times \frac{H}{2^k} \times \frac{W}{2^k} \times 3$. Each clip in the input pyramid is then fed into a weight-shared (*FILM*) encoder to create a set of feature pyramids (Eq. (1)):

$$\left( \{F_0^i\}_{i=1}^d, \{F_1^i\}_{i=1}^d, \ldots, \{F_k^i\}_{i=1}^d \right) = \text{FILM} \left( V_0, V_1, \ldots, V_k \right), \tag{1}$$

$$F = \text{Concat} \left( F_0^d, F_1^{d-1}, \ldots, F_k^1 \right), \tag{2}$$

where $d$ denotes the maximum depth in the feature pyramid. A scale-agnostic feature $F$ is constructed by channel-wise concatenating features from different depths but with the same spatial dimensions (Eq. (2)). We use temporal average pooling to align the dimensions of the features $\{F_1^{d-1}, \ldots, F_k^1\}$ with $F_0^d$ before concatenation, as depicted in Fig. 2(a). By sharing the encoder weights across the input pyramid, the key intuition here is that large motion observed at the deeper depths of $V_0$ should be the same as small motion at the shallower depths of $V_k$, thus, aggregating features from different depths of the feature pyramid allows us to boost the number of pixels available to effectively encode large motion (Reda et al., 2022).

### 2.3 MIDDLE BLOCK

The output of the FILM encoder, as defined in Eq. (2), is fed into the middle block, where the scale-agnostic feature is projected into a latent representation with a reduced channel size. This block is tasked with sampling from the learned distribution of the encoded latent space and generating a latent representation $v$ with reduced spatial and temporal dimensionality. As shown in Fig. 1, the middle block consists of a cascade of *Causal3DResBlock*, *STAttnBlock*, and a Gaussian sampling layer. Following previous works (Kingma & Welling, 2013; Rombach et al., 2022), we adopt an isotropic Gaussian distribution to parameterize the encoded latent variables, from which samples are drawn by the sampling layer.

### 2.4 DECODER

The decoder of our video VAE maps the generated latent back to the input visual data. The network architecture of the decoder resembles that of the base encoder, constructed in a bottom-up manner as shown in Fig. 1. It includes a causal 3D residual block (*Causal3DResBlock*), spatio-temporal upsampling block (*STUpBlock*), and spatio-temporal attention block (*STAttnBlock*).

**Spatio-Temporal Upsampling**  Given a feature volume $x$ of size $b \times (1 + t) \times h \times w \times c$, the *STUpBlock* increases the spatial and temporal dimensions of $x$ by a factor of two, *i.e.* $b \times (1 + 2t) \times 2h \times 2w \times c$. Similar to the downsampling block, the *STUpBlock* is designed as a dual-path module, incorporating both learnable and non-learnable kernels, which are combined through summation (see Fig. 2(b)). We use a causal *transposed* 3D convolutional layer for learnable upsampling. To map $1 + t$ frames to $1 + 2t$ frames, thus enabling the joint usage of our model for both images and videos, we discard the first frame after the upsampling process. A nearest-neighbor interpolation in both spatial and temporal dimension is used for non-learnable upsampling.

### 2.5 NETWORK TRAINING

Following previous works (Rombach et al., 2022; Kingma & Welling, 2013), we train our video VAE using the standard set of loss functions. First, we compute the reconstruction loss $\mathcal{L}_R$ which is calculated as the $L_1$ loss between the input video $V$ and the decoded video $\hat{V}$ (Eq. (3)). We also optimize the perceptual similarity $\mathcal{L}_P$ between each input video frame and the corresponding reconstructed frame using frame-wise LPIPS (Zhang et al., 2018) loss.

$$\mathcal{L}_R = \sum_{i=1}^{1+T} \left| V_i - \hat{V}_i \right| \tag{3}$$

where, $V_i$ denotes the $i^{\text{th}}$ frame index in the input video. To mitigate arbitrary high-variance in the encoded latent spaces, we apply a KL regularization loss $\mathcal{L}_{KL}$ by guiding the learned latent distribution towards a standard normal (Kingma & Welling, 2013; Rombach et al., 2022).

**Flow Regularization**  To ensure that the decoded video accurately preserves the motion dynamics of the input video, we propose a flow regularization loss $\mathcal{L}_{flow}$ for video VAE training. We define $\mathcal{L}_{flow}$ as the mean-squared error between the optical flows of the input video frames and their corresponding optical flows in the decoded video frames. To compute the optical flows, we use pretrained RAFT (Teed & Deng, 2020) model on-the-fly. We employ a bidirectional scheme, as shown in Eq. (4), to ensure robust motion supervision. Our experiments reveal that a model trained with flow regularization decodes temporally smoother motion compared to a model trained without it (refer to Sec. 4). $\mathcal{L}_f$ also facilitates better capturing of motion at higher temporal compression rates.

$$\mathcal{L}_{\text{flow}} = \sum_{i=1}^{T} \left[ \left\| f_{i \to i+1} - \hat{f}_{i \to i+1} \right\|_2^2 + \left\| f_{i+1 \to i} - \hat{f}_{i+1 \to i} \right\|_2^2 \right] \tag{4}$$

where $\hat{f}_{i \to i+1}$ represents the optical flow between $\hat{V}_i$ and $\hat{V}_{i+1}$ frames in the decoded video. The total loss for training our video VAE is formulated as follows:

$$\mathcal{L}_{\text{vae}} = \mathcal{L}_{\text{R}} + \mathcal{L}_{\text{P}} + \alpha_{\text{flow}} \mathcal{L}_{\text{flow}} + \alpha_{\text{KL}} \mathcal{L}_{\text{KL}} \tag{5}$$

where $\alpha_{\text{flow}}$ and $\alpha_{\text{KL}}$ denote the weights for the flow and KL regularization losses, respectively.

**GAN Training** In addition to standard network training, we incorporate adversarial training, following prior works (Esser et al., 2021; Rombach et al., 2022; Blattmann et al., 2023c), to enhance the quality of the decoded video. We optimize a 3D convolution-based PatchGAN discriminator (Isola et al., 2017) to distinguish between real videos and those generated by our video VAE. Our discriminator architecture builds on Pix2Pix (Isola et al., 2017), adapting it for video data by replacing 2D convolutional and batch normalization layers with their 3D counterparts. For further details, please refer to the official Pix2Pix implementation here.

## 3 EXPERIMENT

### 3.1 VIDEO/IMAGE AUTOENCODING

**Implementation Details** We use the WebVid-2M (Bain et al., 2021) dataset for model training. For each step, we randomly sample $T + 1$ consecutive frames from a video, where $T \in \{8, 16\}$, and crop them to a size of $128 \times 128$. The resulting clip with dimensions $(1 + T) \times 128 \times 128 \times 3$ is fed into the video VAE. The FILM encoder uses an input pyramid with $k = 3$. GAN training begins after the initial 100K iterations with $\mathcal{L}_{\text{vae}}$. The flow and KL regularization loss weights are set to $\alpha_{\text{flow}} = 1e - 3$ and $\alpha_{\text{KL}} = 1e - 6$, respectively. We use the Adam (Kingma & Ba, 2014) optimizer with a learning rate of $4.5e - 5$. Training is conducted for 250K iterations with a batch size of 48 on 48 NVIDIA A100 (40GB) GPUs.

**Baseline Methods** We benchmark our approach against state-of-the-art methods for which open-source code or models are available. These include autoencoders from VideoGPT (Yan et al., 2021b), LDM (Rombach et al., 2022), Video LDM (Blattmann et al., 2023c), SVD (Blattmann et al., 2023b), Open-Sora (Zheng et al., 2024), Open-Sora-Plan (Lab & etc., 2024), CV-VAE (Zhao et al., 2024), HVDM (Kim et al., 2024a), and OmniTokenizer (Wang et al., 2024a), CogVideoX (Yang et al., 2024). All models, including ours, encode latents with a channel size of 4 in *continuous* space, except for OmniTokenizer (channel size of 8), CogVideoX (channel size of 16), and VideoGPT, which encodes latents in *discrete* space. For a fair comparison, we also include baselines of our model with latent channel sizes of 8 and 16, as shown in Table 1.

**Evaluation Datasets and Metrics** We evaluate our model and competing approaches on the video autoencoding task using two representative datasets that feature medium to large degrees of motion: Xiph-2K (Niklaus & Liu, 2020) and DAVIS (Pont-Tuset et al., 2017), both at 480p resolution. Additionally, we benchmark image autoencoding performance using the ImageNet validation set (Russakovsky et al., 2015) at $256 \times 256$ resolution. Decoded video/image quality is assessed using PSNR, SSIM, and LPIPS (Zhang et al., 2018), while temporal coherence between frames is assessed using an optical flow-based temporal smoothness (TS) metric (Shen et al., 2020). We also use two video-based metrics: reconstruction FVD (Unterthiner et al., 2019) (rFVD) and reconstruction STREAM (Kim et al., 2024b) (rSTR$_{\text{avg}}$), which focus on the spatio-temporal perceptual quality and fidelity of the decoded videos. For STREAM, we report the average of the fidelity (STREAM-F) and temporal flow (STREAM-T) scores. Please refer to Kim et al. (2024b) for further details.

### 3.1.1 VIDEO COMPRESSION

In Table 1, we comprehensively evaluate our approach and state-of-the-art methods on the video autoencoding task. We train 8 variants of our model with different spatio-temporal compression rates and latent channel sizes. As expected, methods that focus solely on spatial compression, such as Video LDM (Blattmann et al., 2023c), demonstrate strong performance, particularly on videos with large motion. However, as shown in Table 1, our model gives a highly competitive, if not superior, performance compared to competing approaches with significantly higher dimension reduction. For instance, VideoGPT (Yan et al., 2021a) attains an average PSNR of 30.58 dB across the two

Table 1: Quantitative comparison on spatio-temporal video compression

| Method | Comp. Rate | Ch. Size | Xiph-2K | | | | | DAVIS | | | | |
|---|---|---|---|---|---|---|---|---|---|---|---|---|
| | $t \times h \times w$ | $|\hat{z}|$ | PSNR↑ | LPIPS↓ | TS↓ | rFVD↓ | rSTR$_{avg}$↑ | PSNR↑ | LPIPS↓ | TS↓ | rFVD↓ | rSTR$_{avg}$↑ |
| LDM | $1 \times 8 \times 8$ | 4 | 29.72 | 0.136 | -1.41 | 34.26 | 0.839 | 30.36 | 0.136 | -0.83 | 41.63 | 0.785 |
| Video LDM | $1 \times 8 \times 8$ | 4 | 30.32 | 0.120 | -1.76 | 30.46 | 0.924 | 32.30 | 0.131 | -0.81 | 40.10 | 0.889 |
| SVD | $1 \times 8 \times 8$ | 4 | 30.40 | 0.122 | -1.58 | 30.96 | 0.880 | 31.74 | 0.138 | -0.77 | 42.24 | 0.818 |
| HVDM | $1 \times 8 \times 8$ | 4 | 31.16 | 0.114 | -1.36 | 36.55 | 0.827 | 29.63 | 0.170 | -0.42 | 52.04 | 0.777 |
| Ours | $1 \times 8 \times 8$ | 4 | 34.08 | 0.068 | -2.04 | 17.26 | 0.982 | 32.49 | 0.118 | -0.98 | 36.12 | 0.916 |
| VideoGPT | $4 \times 4 \times 4$ | - | 31.04 | 0.225 | -1.09 | 57.11 | 0.762 | 30.12 | 0.306 | 0.14 | 93.67 | 0.760 |
| Ours | $4 \times 4 \times 4$ | 4 | 33.91 | 0.077 | -1.83 | 19.54 | 0.940 | 32.25 | 0.124 | -0.86 | 38.77 | 0.880 |
| Open-Sora | $4 \times 8 \times 8$ | 4 | 28.20 | 0.162 | -1.20 | 42.39 | 0.789 | 26.79 | 0.262 | 0.27 | 80.20 | 0.747 |
| Open-Sora-Plan | $4 \times 8 \times 8$ | 4 | 29.33 | 0.138 | -1.07 | 35.02 | 0.757 | 27.17 | 0.257 | 0.38 | 78.67 | 0.707 |
| CV-VAE | $4 \times 8 \times 8$ | 4 | 29.31 | 0.164 | -1.06 | 41.62 | 0.755 | 29.06 | 0.233 | 0.66 | 71.33 | 0.711 |
| Ours | $4 \times 8 \times 8$ | 4 | 32.12 | 0.107 | -1.53 | 27.16 | 0.868 | 30.31 | 0.159 | -0.24 | 48.67 | 0.806 |
| OmniTokenizer | $4 \times 8 \times 8$ | 8 | 28.61 | 0.153 | -1.05 | 38.83 | 0.753 | 26.86 | 0.260 | 0.41 | 79.59 | 0.712 |
| CogVideoX | $4 \times 8 \times 8$ | 16 | 32.78 | 0.071 | -1.51 | 18.02 | 0.913 | 31.07 | 0.124 | -0.50 | 37.96 | 0.852 |
| Ours | $4 \times 8 \times 8$ | 8 | 33.39 | 0.084 | -1.70 | 21.32 | 0.909 | 31.65 | 0.128 | -0.54 | 39.18 | 0.855 |
| Ours | $4 \times 8 \times 8$ | 16 | 34.51 | 0.062 | -1.88 | 15.74 | 0.952 | 32.86 | 0.104 | -0.86 | 31.84 | 0.892 |
| Ours | $8 \times 8 \times 8$ | 4 | 31.28 | 0.121 | -1.37 | 30.71 | 0.829 | 29.04 | 0.226 | 0.16 | 69.18 | 0.786 |
| Ours | $16 \times 8 \times 8$ | 4 | 29.74 | 0.162 | -0.96 | 41.12 | 0.731 | 27.51 | 0.270 | 0.39 | 82.65 | 0.679 |
| Ours | $16 \times 16 \times 16$ | 4 | 26.91 | 0.250 | -0.76 | 63.45 | 0.683 | 24.60 | 0.356 | 0.47 | 108.98 | 0.644 |

Figure 3: Qualitative comparison of decoded video frames between our model and competing approaches

datasets at a compression rate of $4 \times 4 \times 4$. In comparison, our model achieves 30.16 dB at a compression rate of $8 \times 8 \times 8$. This can be attributed to the proposed spatio-temporal downsampling and upsampling modules, which effectively leverage both learnable and non-learnable kernels to facilitate the encoding and decoding of information at higher compression rates. Additionally, the use of the FILM encoder mitigates the challenges of encoding large motion, contributing to the strong performance of our video VAE on large-motion datasets (Niklaus & Liu, 2020; Pont-Tuset et al., 2017). It can also be observed that our $16 \times 8 \times 8$ model achieves performance comparable to the best competing method at a $4 \times 8 \times 8$ compression rate, Open-Sora-Plan (Lab & etc., 2024). The proposed flow regularization loss enables faithful reconstruction of motion between decoded frames, resulting in robust autoencoding performance at higher temporal compression, as evident in Table 1.

In Fig. 3, we qualitatively compare our model with the best-performing baselines. As shown in the figure, our $8 \times 8 \times 8$ model successfully reconstructs a small, fast-moving object (highlighted in the green box), whereas Open-Sora-Plan fails at a $4 \times 8 \times 8$ compression rate. It is also evident that our approach reconstructs human faces with great fidelity, while other methods with much lower compression rates, such as VideoGPT, struggle to preserve details (see the *nose* and *mouth* in the blue box). As illustrated in Fig. 3, our model generally reconstructs sharper frames while maintaining the structural details of objects that are far from the camera, compared to other methods (see the details in the red box). Please refer to the supplemental video and the appendix for further analyses.

### 3.1.2 IMAGE COMPRESSION

We evaluate our approach and other baselines on the image autoencoding task, as shown in Table 2. As can be inferred from the table, our causal video VAE demonstrates competitive image compression performance across various metrics. Notably, our model, trained exclusively on video clips without explicit optimization for images, performs comparably to a state-of-the-art image VAE (Rombach et al., 2022). These results highlight the advantages of the causal formulation in our video VAE for a joint image and video compression model.

## 3.2 VIDEO GENERATION

Our work has practical applications in tasks such as image/video generation, where the encoder compresses visual data into a latent space for training generative models, and during inference, the

Table 2: Comparison on image compression

| Method | $t \times h \times w$ | $|\hat{z}|$ | PSNR ↑ | SSIM ↑ | LPIPS ↓ |
|---|---|---|---|---|---|
| | | | **ImageNet Val** | | |
| LDM | $1 \times 8 \times 8$ | 4 | 29.06 | 0.684 | 0.137 |
| Open-Sora | $4 \times 8 \times 8$ | 4 | 26.67 | 0.692 | 0.161 |
| Open-Sora-Plan | $4 \times 8 \times 8$ | 4 | 27.57 | 0.654 | 0.145 |
| OmniTokenizer | $4 \times 8 \times 8$ | 4 | 27.05 | 0.651 | 0.150 |
| CV-VAE | $4 \times 8 \times 8$ | 4 | 27.69 | 0.652 | 0.141 |
| Ours | $4 \times 8 \times 8$ | 4 | 29.02 | 0.697 | 0.088 |
| CogVideoX | $4 \times 8 \times 8$ | 16 | 29.66 | 0.712 | 0.075 |
| Ours | $4 \times 8 \times 8$ | 16 | 30.54 | 0.744 | 0.056 |

Table 3: Comparison on video generation

| Method | SkyTimelapse | | UCF-101 | |
|---|---|---|---|---|
| | $\text{FVD}_{16} \downarrow$ | $\text{FVD}_{128} \downarrow$ | $\text{FVD}_{16} \downarrow$ | $\text{FVD}_{128} \downarrow$ |
| VideoGPT | 222.7 | - | 2880.6 | - |
| DIGAN | 83.11 | 196.7 | 1630.2 | 2293.7 |
| StyleGAN-V | 79.52 | 197.0 | 1431.0 | 1773.4 |
| PVDM | 71.46 | 159.9 | 457.4 | 902.2 |
| LDM + Open-Sora | 53.38 | 127.5 | 266.8 | 657.6 |
| Open-Sora | 55.12 | 130.9 | 283.4 | 662.9 |
| Ours + Open-Sora | **49.50** | **119.8** | **248.7** | **572.1** |

decoder reconstructs the data from the generated latents (Brooks et al., 2024). To demonstrate this, we plug in our pretrained video VAE into the diffusion-based open-source generative framework, Open-Sora (Zheng et al., 2024), and train it for unconditional video generation.

**Implementation Details**    We use the train-split of commonly used video synthesis benchmarks, SkyTimelapse (Zhang et al., 2020) and UCF-101 (Soomro, 2012), for model training. Consistent with previous works (Yu et al., 2023b; Skorokhodov et al., 2022), we resize each frame in the datasets to a resolution of $256 \times 256$ and sample video clips of length $1 + T$, where $T \in \{16, 128\}$. Our video generation experiments are conducted on 16 NVIDIA A100 (80GB) GPUs adhering to the training configuration in Zheng et al. (2024).

**Baseline Methods**    We compare our video VAE ($4 \times 8 \times 8$) with the autoencoders from LDM ($1 \times 8 \times 8$) and Open-Sora ($4 \times 8 \times 8$) on the video generation task by integrating them into a video generation model (Zheng et al., 2024). All autoencoders, including ours and the baselines, are kept frozen during the generation experiments. Additionally, we report the results of state-of-the-art video generation methods, including VideoGPT (Yan et al., 2021a), DIGAN (Yu et al., 2022), StyleGAN-V (Skorokhodov et al., 2022), and PVDM (Yu et al., 2023b).

**Evaluation**    We use the Fréchet Video Distance (FVD) (Unterthiner et al., 2018) to quantitatively assess the quality of the generated clips. Following the evaluation protocol in previous works (Yu et al., 2023b; Skorokhodov et al., 2022), we measure the FVD score on video clip lengths of 16 and 128 frames. For both $\text{FVD}_{16}$ and $\text{FVD}_{128}$, we evaluate using 2,048 real and generated video clips.

### 3.2.1 RESULTS

In Table 3, we present a quantitative comparison of Open-Sora (Zheng et al., 2024) (which utilizes latents encoded by LDM (Rombach et al., 2022), Open-Sora's video VAE, and our video VAE) and previous video generation methods. The significant performance improvement of Open-Sora-based generation over prior GAN-based (Yu et al., 2022; Skorokhodov et al., 2022) and autoregressive-based (Yan et al., 2021a) approaches, as evident from the table, can be attributed to the state-of-the-art diffusion-based architecture in Zheng et al. (2024). It can also be inferred that using our video VAE (*Ours + Open-Sora*) delivers the best performance in both short and long video generation, notably outperforming Open-Sora's own video VAE. Additionally, our video VAE achieves better generation results than LDM, as shown in Table 3, while being $4\times$ more efficient in temporal compression. These results highlight the effectiveness of our autoencoder in generative modeling.

In Fig. 4, we visualize sequence of video frames generated using our video VAE. The first three rows illustrate generated results based on the SkyTimelapse dataset, while the last three rows show generated results from the UCF-101 dataset. As shown in the figure, Open-Sora, powered by our video VAE, generates realistic sky time-lapse videos with high fidelity. It can also be seen that our method, trained solely on the challenging UCF-101 dataset, synthesizes videos of human faces with satisfactory details and videos of human actions exhibiting spatio-temporal consistency, as depicted in Fig. 4.

## 4    ABLATION STUDIES

We conduct extensive ablation studies to analyze the contributions of the different components in the proposed video VAE, including the FILM encoder, the spatio-temporal down/upsampling blocks, the spatio-temporal attention block, and the flow regularization loss. All experiments are conducted on a video VAE with a spatio-temporal compression rate of $8 \times 8 \times 8$. The results on the Xiph-2K (Niklaus & Liu, 2020) and DAVIS (Pont-Tuset et al., 2017) datasets are summarized in Table 4.

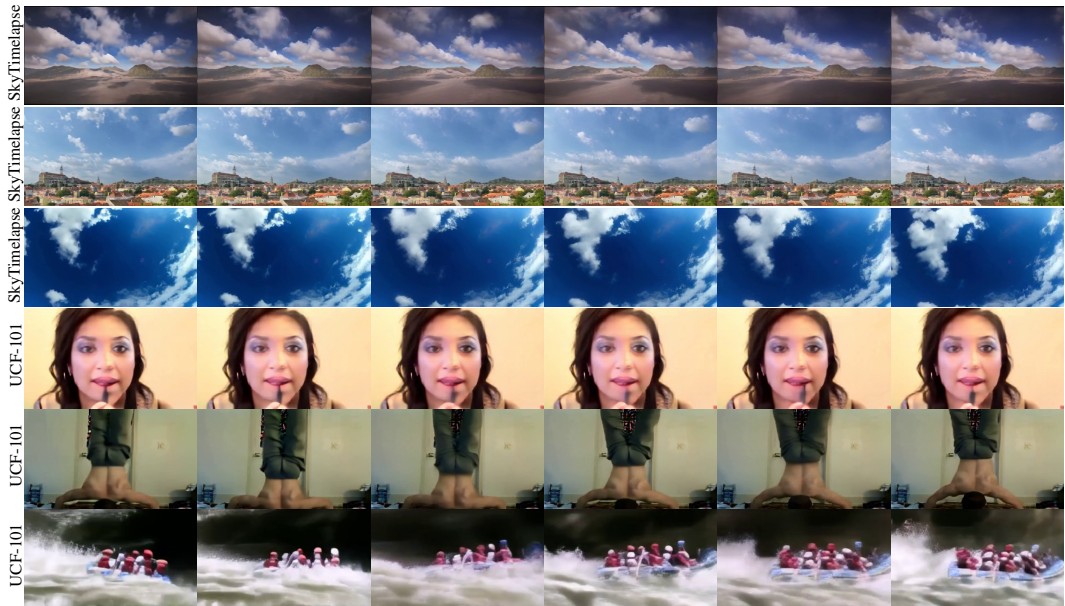

Figure 4: Qualitative analysis of videos generated by Open-Sora, enabled by our video VAE (*Ours + Open-Sora*). The top three rows are from a model trained on SkyTimelapse, and the bottom three on UCF-101.

**FILM Encoder**    We explore the advantages of employing a weight-shared encoder across the input video pyramid to address the challenges associated with decoding small and fast-moving objects, as outlined in Sec. 2.2. To accomplish this, we train our video VAE using a single (base) encoder that takes the original resolution of the input video and compare its performance with a model that uses a FILM encoder. As can be inferred from Table 4(a), a model with a FILM encoder consistently outperforms a single encoder model across all metrics. Specifically, on the DAVIS dataset, which predominantly contains large motion, using a FILM encoder improves performance by 1.81 dB on the PSNR metric. The qualitative results in Fig. 5 further show that a video VAE with a FILM encoder reconstructs sharper frames compared to one without it. These findings underscore the advantage of learning scale-agnostic features through shared encoder weights, which helps mitigate the challenges of large-motion decoding (Reda et al., 2022).

**Spatio-Temporal Down/Upsampling**    Here, we study how different network designs for the down/upsampling modules impact our video VAE. First, we train a model using *non-learnable kernels* (followed by a convolution layer), employing average pooling for downsampling and nearest interpolation for upsampling, in line with previous works (Rombach et al., 2022; Lab & etc., 2024). Additionally, we train a separate model using *learnable kernels* (Yu et al., 2024), employing strided convolution for downsampling and strided transposed convolution for upsampling. We compare these baselines against our proposed approach, which integrates both learnable and non-learnable kernels, as described in Sec. 2. The results for input videos sampled at different sequence lengths are presented in Table 4(b). As can be inferred from the table, a model based on only learnable kernels outperforms a model based on non-learnable kernels for a sequence length seen during training ($T = 8$). However, when tested at a different sequence length ($T = 32$), the performance of the model based on learnable kernels significantly drops compared to the model based on non-learnable kernels. We observe that while spatio-temporal down/upsampling with 3D convolutions generally performs well, the model tends to overfit to the training sequence length. This results in a notably worse performance at different sequence lengths during inference. In comparison, non-learnable spatio-temporal down/upsampling in a video VAE results in relatively lower performance but is more robust across different sequence lengths, as seen in Table 4(b). Our proposed method which combines the best of both worlds as depicted in Fig. 2(b), not only outperforms both approaches but is also temporally more adaptable to different sequence lengths.

**Flow Regularization**    To examine the benefit of imposing motion constraints in video VAE training, we train our model with and without the flow regularization loss (Eq. (4)). The results are presented in Table 4(c). As evident from the table, a model trained with flow regularization loss $\mathcal{L}_{\text{flow}}$ consistently outperforms a model trained without it. In particular, there is a significant improvement in the

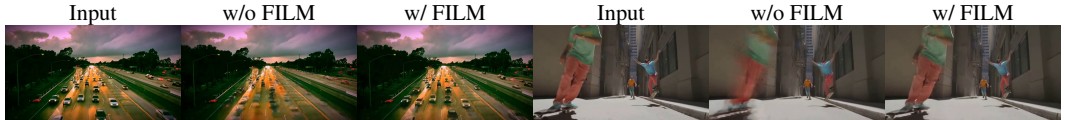

| Input | w/o FILM | w/ FILM | Input | w/o FILM | w/ FILM |

Figure 5: Qualitative analysis of large-motion decoding with (w/) and without (w/o) the FILM encoder

Table 4: Ablation experiments on different components of our causal video VAE

| | Method | Xiph-2K | | | | DAVIS | | | |
|---|---|---|---|---|---|---|---|---|---|
| | | PSNR ↑ | SSIM ↑ | LPIPS ↓ | TS ↓ | PSNR ↑ | SSIM ↑ | LPIPS ↓ | TS ↓ |
| (a) | w/o FILM encoder | 30.44 | 0.775 | 0.139 | -0.86 | 27.23 | 0.587 | 0.251 | 0.29 |
| | w/ FILM encoder | 31.28 | 0.797 | 0.121 | -1.37 | 29.04 | 0.624 | 0.226 | 0.16 |
| (b) | non-learnable kernel ($T = 8$) | 29.98 | 0.728 | 0.156 | -1.13 | 27.89 | 0.560 | 0.257 | 0.21 |
| | non-learnable kernel ($T = 32$) | 29.60 ($\downarrow 0.38$) | 0.710 | 0.175 | -1.08 | 27.47 ($\downarrow 0.42$) | 0.524 | 0.276 | 0.26 |
| | learnable kernel ($T = 8$) | 30.60 | 0.743 | 0.153 | -1.18 | 28.03 | 0.555 | 0.257 | 0.18 |
| | learnable kernel ($T = 32$) | 28.98 ($\downarrow 1.62$) | 0.713 | 0.182 | -1.01 | 26.95 ($\downarrow 1.08$) | 0.489 | 0.305 | 0.33 |
| | ours ($T = 8$) | 31.28 | 0.797 | 0.121 | -1.37 | 29.04 | 0.624 | 0.226 | 0.16 |
| | ours ($T = 32$) | 30.85 ($\downarrow 0.43$) | 0.785 | 0.130 | -1.25 | 28.82 ($\downarrow 0.22$) | 0.619 | 0.234 | 0.17 |
| (c) | w/o flow regularization ($\mathcal{L}_{\text{flow}}$) | 30.65 | 0.784 | 0.130 | -0.68 | 28.26 | 0.607 | 0.236 | 0.58 |
| | w/ flow regularization ($\mathcal{L}_{\text{flow}}$) | 31.28 | 0.797 | 0.121 | -1.37 | 29.04 | 0.624 | 0.226 | 0.16 |
| (d) | w/o *STAttnBlock* | 30.52 | 0.773 | 0.139 | -0.83 | 27.36 | 0.603 | 0.240 | 0.36 |
| | w/ *STAttnBlock* | 31.28 | 0.797 | 0.121 | -1.37 | 29.04 | 0.624 | 0.226 | 0.16 |

temporal smoothness of the decoded frames, as noted from the TS metric in Table 4(c). This implies that incorporating flow regularization during training is advantageous for decoding temporally consistent videos. Please refer to the supplemental video for qualitative examples.

**Spatio-Temporal Attention** Here, we analyze the importance of incorporating self-attention layers (Vaswani et al., 2017) into our video VAE to explicitly capture spatio-temporal dependencies. To achieve this, we train our model after removing all the attention blocks from the encoder, middle block, and decoder parts of our network (see Fig. 1). As can be observed from Table 4(d), a model trained without *STAttnBlock* already gives a competitive performance, as causal 3D convolutional layers are strong in capturing spatio-temporal correspondences in videos (Yu et al., 2024). However, adding spatio-temporal attention blocks in the video VAE results in a notable performance boost.

## 5 CONCLUSION

This work presents a causal video VAE for high-quality video compression. We make three key contributions. First, we propose a weight-shared encoder that efficiently captures multi-scale features from the input video. This mitigates the challenges associated with reconstructing videos with large motions. Second, we introduce robust spatio-temporal down/upsampling blocks, overcoming the limitations of prior methods. Third, to preserve motion dynamics during high compression, we introduce a flow regularization loss for video VAE training. We demonstrate the effectiveness of our model through comprehensive experiments. We also demonstrate the potential of our proposed model as a robust autoencoder for video generation training.

## ACKNOWLEDGMENT

This work was partially supported by IITP grant funded by the Korea government (MSIT) (RS-2024-00457882, National AI Research Lab Project).

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

## A   APPENDIX

## B   RELATED WORKS

**Video Compression**   Traditionally, video compression has relied on handcrafted algorithms such as H.264 (Wiegand et al., 2003) and H.265 (Sullivan et al., 2012). These algorithms achieve excellent compression rates but require video coding experts to meticulously design and optimize the various components of the compression pipeline. In contrast, neural networks that can automatically learn important features during training have emerged as strong contenders (Mentzer et al., 2022; Djelouah et al., 2019; Rippel et al., 2021; Lin et al., 2020). Pioneering works such as DVC (Lu et al., 2019) mimic traditional compression methods, replacing certain components within the framework with neural networks. Following works (Agustsson et al., 2020; Lin et al., 2020; Djelouah et al., 2019; Rippel et al., 2021) build upon this framework, introducing enhancements to various modules. Branching away from the complex residual coding-based framework which consists of many modules, some works (Habibian et al., 2019; Mentzer et al., 2022; Hu et al., 2023b) opt for a simpler autoencoder-based approach. For instance, Habibian et al. (2019) uses a 3D autoencoder and an auto-regressive prior coding model, Hu et al. (2023b) combines previous modules for compression in feature space using an autoencoder-style network, and  Mentzer et al. (2022) utilizes image autoencoders to obtain frame-wise representations of videos, which are then quantized through a trained Transformer (Vaswani et al., 2017). Except for a few works (Hu et al., 2023b; Mentzer et al., 2022; Habibian et al., 2019) that perform dimension reduction, although not significantly, the primary focus of neural compression has been on reducing bitrate rather than the dimension.

**Video Compression for Generation**   In recent years, generative modelling (Esser et al., 2021; Yu et al., 2024; Rombach et al., 2022; Yu et al., 2023a; Chang et al., 2022) has ushered in a new era of neural video compression approaches, with a primary focus on achieving low-dimensional latent representations. The emphasis on obtaining low-dimensional latents is because highly compressed latents allow generative models to ignore imperceptible pixel variations, providing an efficient generation environment and reducing computational costs. These approaches can broadly be divided into those that learn *discrete* latent representations and those that learn *continuous* latent representations. Discrete neural video compression approaches mainly follow a pioneering work (van den Oord et al., 2018), which uses a learned codebook through the k-means clustering algorithm for image compression. For example, Yan et al. (2021b) follows a similar approach but uses 3D convolutions to properly model spatio-temporal dynamics for effective video compression. Recently, follow-up works (Yu et al., 2024; Mentzer et al., 2023) have focused on non-learnable and look-up free quantization schemes instead of the k-means clustering to learn a bigger codebook for more faithful reconstructions.

On the other hand, continuous neural video compression approaches follow after the VAE (Kingma & Welling, 2013) framework, incorporating perceptual losses and GAN training to enhance perceptual quality, as introduced in Rombach et al. (2022) for image compression. Notably, techniques such as Zeng et al. (2023); Qing et al. (2023); Xing et al. (2023); Jain et al. (2024); Hu et al. (2023a) utilize a pretrained image autoencoder for independent spatial compression of video frames. However, this approach may induce flickering artifacts (Blattmann et al., 2023c). To address this issue, Blattmann et al. (2023c;a) fine-tune a pretrained image autoencoder with added 3D convolutions to align latents for temporal reasoning, employing a training framework adapted for videos. Another noteworthy approach is Yu et al. (2023c), which performs spatial compression by learning a triplane low-dimensional representation of the video. However, these approaches neglect to exploit the well-documented temporal redundancies in videos (Lu et al., 2019; Lin et al., 2020; Hu et al., 2023b), resulting in larger latent dimensions. To address these limitations, our work leverages temporal redundancies and proposes a continuous video autoencoder that performs compression in both the spatial and temporal domains. We also empirically demonstrate the potential of our proposed model as a robust autoencoder for video generation training.

### B.1   EXPERIMENTAL ANALYSES

### B.1.1   LATENT SENSITIVITY

In this analysis, we investigate the robustness of the video VAE decoder to erroneous latent predictions from a generative model. To do so, we intentionally corrupt the encoded latents using various corruption methods and assess the quality of the video reconstructed by the decoder. First, we use a

Table 5: Experimental analyses on latent sensitivity and arbitrary sequence length on Adobe240 dataset

| Method | Spatio-Temporal Compression $t \times h \times w$ | Noise $\sigma = 0$ | Noise $\sigma = 0.4$ | Noise $\sigma = 0.8$ | Quantization fp32 | Quantization bf16 | Quantization uint8 | Sequence Length $T = 8$ | Sequence Length $T = 64$ | Sequence Length $T = 128$ |
|---|---|---|---|---|---|---|---|---|---|---|
| LDM | $1 \times 8 \times 8$ | 28.53 | 27.86 | 26.91 | 28.53 | 28.41 | 15.56 | 28.53 | 28.53 | 28.53 |
| Video LDM | $1 \times 8 \times 8$ | 29.51 | 28.30 | 27.02 | 29.51 | 29.49 | 16.50 | 29.51 | 29.16 | 29.04 |
| Open-Sora-Plan | $4 \times 8 \times 8$ | 28.43 | 27.31 | 25.67 | 28.43 | 28.21 | 16.11 | 28.43 | 27.92 | 27.34 |
| Ours | $8 \times 8 \times 8$ | 31.50 | 30.96 | 29.03 | 31.50 | 31.42 | 17.49 | 31.50 | 31.22 | 30.83 |
| Ours ($T = 64$) | $8 \times 8 \times 8$ | 31.22 | 30.89 | 28.92 | 31.22 | 31.20 | 17.16 | - | - | - |

| Input | $\sigma = 0, fp32$ | $\sigma = 0.4$ | $\sigma = 0.8$ | bf16 | uint8 |
|---|---|---|---|---|---|

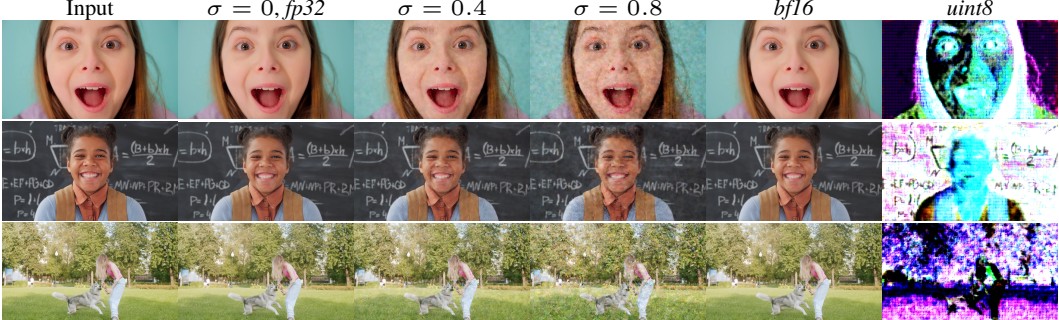

Figure 6: Qualitative analysis on the effect of noise corruption and quantization on encoded latents

noise corruption scheme where we add different magnitudes of Gaussian noise (of the same size as the encoded latent) to the encoded latent *i.e.* $\hat{v} = v + \sigma \cdot \mathcal{N}(0, 1)$, where $v$ denotes the encoded latent. We experiment with 3 scales of noise: $\sigma = 0$ (clean), $\sigma = 0.4$ (medium noise) and $\sigma = 0.8$ (large noise). As anticipated, the decoded video quality decreases in proportion to the scale of the added Gaussian noise as shown in Table 5. However, our video decoder still performs reasonably well even when the latents are corrupted with significant noise. A similar conclusion can be drawn from the visualization in Fig. 6, where the overall structure of the video is well-maintained for $\sigma = 0.8$ despite the noisy output. It is also noteworthy from Table 5 that our video VAE exhibits relatively better robustness to noise for longer input sequence lengths. For example, there is a 0.33 dB drop for $\sigma = 0.4$ when $T = 64$ compared to a 0.54 dB drop when $T = 8$. This result further underscores the importance of temporally agnostic video VAEs. Please refer to the supplemental video for qualitative examples.

We also investigate the sensitivity of a video VAE to latent quantization. We achieve this by storing the encoded latents in various data type formats: 32-bit floating point (*fp32*), 16-bit brain floating point (*bf16*), and unsigned 8-bit integer (*uint8*). As demonstrated in Table 5, *bf16* quantization performs comparably to *fp32* despite the reduced precision, while *uint8* quantization leads to poor decoding performance across all methods. This observation is further supported by Fig. 6, where our model with *bf16* decodes video frames indistinguishable from those generated with *fp32*, whereas *uint8* only preserves the edges of the frame. This failure occurs because *uint8* significantly shifts the distribution of the encoded latent, which is optimized to resemble a standard normal distribution via KL regularization (refer to Sec. 2.5).

### B.1.2 SEQUENCE LENGTH

The applicability of a video VAE hinges on its ability to robustly encode and decode arbitrarily long videos sampled at varying lengths. In Table 5, we assess different approaches in this regard using three different sequence lengths: $T = 8$, $T = 64$, and $T = 128$. As expected, methods that only perform spatial compression (Blattmann et al., 2023c) tend to perform consistently across different sequence lengths compared to those that perform both spatial and temporal compression (Lab & etc., 2024). For example, Video LDM (Blattmann et al., 2023c) ($1\times$ in the temporal dimension) shows a 0.47 dB performance drop when the sampling rate changes from $T = 8$ to $T = 128$, while Open-Sora-Plan (Lab & etc., 2024) ($4\times$) exhibits a larger drop of 1.09 dB under the same conditions. In comparison, our method ($8\times$) experiences a 0.67 dB performance drop. These results show that our model, even with a higher compression rate, generalizes reasonably well to varying sequence lengths.

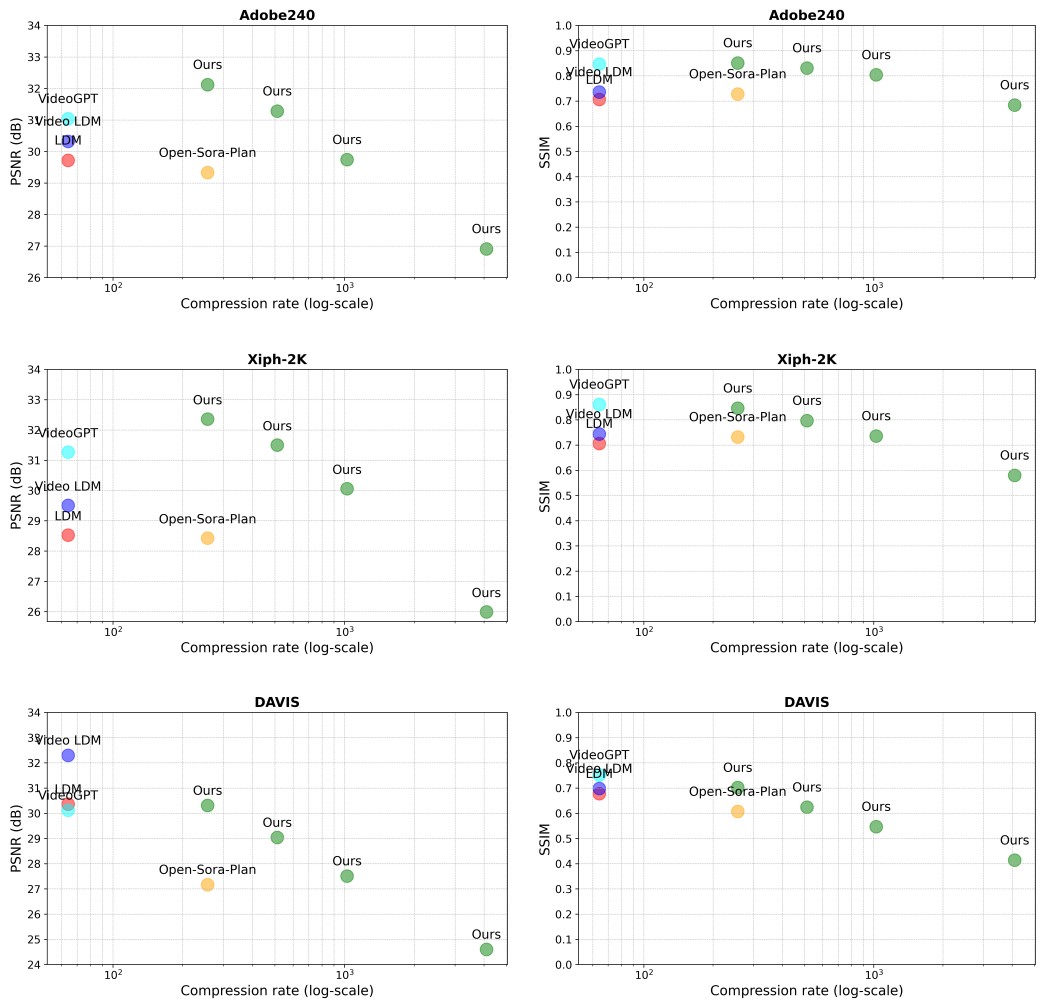

Figure 7: PSNR and SSIM analysis across spatio-temporal compression rates

Table 6: Effect of padding in causal 3D convolution

| Method | PSNR | SSIM | LPIPS | TS |
|---|---|---|---|---|
| zero padding | 28.77 | 0.619 | 0.232 | 0.27 |
| constant padding | 28.86 | 0.620 | 0.231 | 0.22 |
| replication padding | 29.04 | 0.624 | 0.226 | 0.16 |

## B.2 VIDEO COMPRESSION COMPARISON

Here, we present visualizations of the results summarized in Table 1 of our paper. In Fig. 7, we illustrate the PSNR and SSIM (on the y-axis) against the spatio-temporal compression rate of our model and competing approaches on the Adobe240 (Su et al., 2017), DAVIS (Pont-Tuset et al., 2017), and Xiph-2K (Niklaus & Liu, 2020) datasets. As evident from the plots, our model achieves highly competitive, if not superior, performance compared to competing approaches while accomplishing significantly higher dimension reduction.

## B.3 PADDING IN CAUSAL 3D CONVOLUTION

Here, we explore the impact of various padding schemes in the causal 3D convolutional layers within our video VAE. We conduct experiments using three different temporal padding schemes: zero, constant, and replication. Training our $8 \times 8 \times 8$ model with each padding scheme, we evaluate their performance on the DAVIS dataset (Pont-Tuset et al., 2017). The results, outlined in Table 6, indicate that replication padding yields the best performance. Moreover, we observe that replication padding leads to temporally smoother videos compared to zero or constant padding.

