# OpenReview forum: "High-Quality Joint Image and Video Tokenization with Causal VAE"
_ICLR.cc/2025/Conference — ICLR 2025 Poster_

### Official Review · Reviewer_ea5M · 2024-11-02

**Soundness:** 3
**Presentation:** 3
**Contribution:** 2
**Rating:** 6
**Confidence:** 3

**Summary:**

This paper presents valuable improvements in the video VAE domain. The main innovations include a multiple-scales encoder, dual-path spatio-temporal sampling module, and flow regularization loss. The experimental design is comprehensive, particularly in downstream task validation.

**Strengths:**

This paper presents valuable improvements in the video VAE domain. The main innovations include a multiple-scales encoder, dual-path spatio-temporal sampling module, and flow regularization loss. The experimental design is comprehensive, particularly in downstream task validation.

**Weaknesses:**

If the following areas can be optimized, the quality of the paper will improve. Key areas for enhancement include:

1.Enhanced Visualization in Experiments
Current Observation: While the experiments are comprehensive, covering downstream tasks such as image and video generation, this section primarily relies on metric comparisons and lacks crucial visualization analysis.
Recommendations:
Figure 4 Enhancement: Add a comparison of videos generated by OpenSora and the combination of your method with OpenSora. This will provide a clearer visual demonstration of the improvements introduced by your approach.
Ablation Study Visualization: In Figure 5, include visual results showcasing the effects of individual components. For example, display results with and without Spatio-Temporal Down/Upsampling, with and without flow regularization, and with and without Spatio-Temporal Attention. This will help illustrate the contribution of each component to the overall performance.

2.Comparison with SVD-VAE
Current Observation: The paper lacks a comparison with SVD's Variational Autoencoder (SVD-VAE), which is relevant to the presented work.
Recommendation: Incorporate a comparison with SVD-VAE in Table 2. This will provide a more comprehensive evaluation of your method against existing approaches and highlight its relative strengths and weaknesses.

3.Innovation of Improvement Methods
Current Observation: The three proposed improvement methods in the paper are generally intuitive but lack a certain degree of innovation.
Recommendation: To enhance the paper's contribution, consider elaborating on the novelty of each improvement method. Provide deeper insights into how these methods advance the current state of the art beyond intuitive enhancements.

**Questions:**

1. Clarification of the Ablation Study
   How can the ablation study section be enhanced to clearly demonstrate the impact of adding or removing individual metrics? Additionally, would it be beneficial to include comparisons for configurations such as baseline + a, baseline + b, baseline + c, baseline + d, and the combined baseline + a + b + c + d?

2. Inclusion of Visualization Analysis in Experiments
   While the experiments are thorough and encompass downstream tasks like image and video generation with metric comparisons, the analysis lacks essential visualization components. Could the authors incorporate visualization analyses to complement the metric-based evaluations?

3. Comparison with SVD-VAE in Table 1
   I recommend adding a comparison with SVD-VAE in Table 1 to provide a more comprehensive evaluation of the proposed method against existing models.

These refined questions and suggestions aim to provide constructive feedback that can help clarify ambiguities, address potential limitations, and enhance the overall quality of the work during the rebuttal and discussion phases.

---

> ### Author Response · Authors · 2024-11-21
> **Response to Reviewer ea5M**
>
> Thank you for your constructive review and insightful suggestions. We have thoughtfully considered your feedback and revised our paper accordingly, as outlined below.
>
> - **Clarification of the Ablation Study:**
>
>     Following the reviewer’s suggestion, we have included detailed ablation studies in the revised manuscript, analyzing the contribution of each individual component by incrementally incorporating them into a *base model*. The results are summarized in **Table 8** in the appendix, with a related discussion provided in **Appendix A.7**.
>
>     We have also included qualitative visualizations for the ablation experiments in the **supplemental video**.
>
>
> - **Inclusion of Visualization Analysis:**
>
>     We have added qualitative examples of videos generated by Open-Sora using *Open-Sora’s own video VAE* in comparison with using *our video VAE* in the **supplemental video** to strengthen the qualitative analysis of our work. Please refer to the updated **supplemental video**.
>
> - **Comparison with SVD-VAE:**
>
>     Thank you for the suggestion. We have added SVD-VAE in **Table 1** in our revised manuscript. *We would like to clarify that SVD-VAE is not temporally causal and cannot be used for image autoencoding. Therefore, it was not included in Table 2.*

---

> > ### Author Response · Authors · 2024-11-26
> > **Looking Forward to Your Feedback**
> >
> > Dear Reviewer ea5M,
> >
> > Thank you once again for your constructive review and valuable suggestions. We have carefully considered your feedback and made the corresponding updates to our paper, as outlined in our response above. We are reaching out to confirm if there are any remaining concerns we can address **before the discussion period concludes**. We look forward to hearing from you.

---

> > > ### Author Response · Authors · 2024-11-30
> > > **Follow-Up: Looking Forward to Your Feedback**
> > >
> > > Dear Reviewer ea5M,
> > >
> > > With the **author-reviewer discussion period nearing its conclusion**, we kindly request your feedback on whether all your concerns have been fully addressed. Should you have any additional questions or require further clarification, please do not hesitate to let us know. We look forward to hearing from you.
> > >
> > > Paper 2418 Authors

---

### Official Review · Reviewer_MHdT · 2024-11-03

**Soundness:** 3
**Presentation:** 3
**Contribution:** 3
**Rating:** 6
**Confidence:** 4

**Summary:**

This paper presents several improvements to the Video VAE architecture. First, it introduces 3D causal convolutions to encode images and videos simultaneously. Second, it employs a 2 + 1D attention layer for separate interactions of spatiotemporal information, referred to as the STAttnBlock in the paper. Third, it utilizes a two-pathway approach in the upsampling module, with one pathway being learnable and the other non-learnable. Lastly, an interesting contribution is the introduction of the FILM encoder, which shares encoder weights to handle videos resized to different scales. The experiments claim that the proposed method achieves significant results. However, I think that important parameters are not presented in the experiments, and these parameters could greatly affect the experimental metrics.

**Strengths:**

1. The proposed FILM Encoder is designed well: its pyramid structure enhances the model's ability to reconstruct targets of varying sizes, improving detail retention and overall reconstruction quality. The use of shared modules also bolsters the model's robustness. Additionally, the ablation experiments presented in the paper demonstrate that the FILM Encoder indeed improves reconstruction performance.
2. The Flow Regularization introduced in this paper will contribute positively to motion continuity in video reconstruction.
3. The experiments conducted on video generation effectively demonstrate that this VAE structure can be utilized for downstream diffusion training.

**Weaknesses:**

Although the model is well designed, I am skeptical about the correctness of the experiment. In previous work (such as the SD3 report and the CogVideo X report), the number of latent channels has been emphasized as having a significant impact on reconstruction performance and metrics. The CogVideo report demonstrates that a model with a configuration of 4x8x8 and 8 latent channels achieves about 3 points higher PSNR compared to a model with 32 latent channels. However, in the experiments of this paper, the 4x8x8 model shows nearly 4 points higher PSNR than the compression rate model, while the 8x8x8 compression rate model surpasses the PSNR of the 4x4x4 token compression rate VideoGPT by 0.23 points, despite having a 64 times higher compression rate. Even in the 16x8x8 model, the LPIPS metric remains superior to VideoGPT, which is quite surprising. The paper does not specify the number of latent channels in the model, raising concerns about the comparability with similar models (e.g., Open-Sora, which has only 4 latent channels). I believe that the overall compression rate (considering latent channels) is a fundamental factor limiting VAE reconstruction performance. If the total compression rates were the same, the differences in metrics would likely not be so pronounced. Therefore, I am concerned about the rigor of the experiments presented.

**Questions:**

1. Has the author explored the number of latent channels? If so, I would like to see those findings and understand why they were not presented in the paper.
2. How can the authors explain the superior performance of the 8x8x8 compression rate model over the 4x4x4 compression rate model, given the 64 times difference in compression ratio? Similarly, the 16x8x8 model outperforms the 4x4x4 model in LPIPS metrics, despite a 128 times difference in compression ratio. I believe this discrepancy cannot be solely attributed to training tricks or model design.

---

> ### Author Response · Authors · 2024-11-21
> **Response to Reviewer MHdT**
>
> Dear Reviewer MHdT,
>
> Thank you for your valuable feedback. We have revised our manuscript to address your concerns and clarify any potential misunderstandings, as outlined below.
>
> ## Clarification
> 1. We would like to clarify that Section 3.1 (**Baseline Methods**) specifies that our models use a latent channel size of 4. For additional details, please refer to this section.
>
> 2. We would also like to clarify that the **compression ratio difference** between an $8 \times 8 \times 8$ model and a $4 \times 4 \times 4$ model is only $8$ times, *not $64$ times as suggested by Reviewer MHdT*.
>
>     Similarly, the compression ratio difference between a $16 \times 8 \times 8$ model and a $4 \times 4 \times 4$ model is $16$ times, *not $128$ times as suggested by Reviewer MHdT*.
>
> ## Response to questions
>
> 1.  **Has the author explored the number of latent channels? If so, I would like to see those findings and understand why they were not presented in the paper.**
>
>     - We explored latent channel sizes of $4$, $8$, and $16$. Our experimental observations indicate that, for the same spatio-temporal compression rate, higher latent channel sizes consistently improve autoencoding performance ($16 > 8 > 4$), which aligns with the findings in CogVideoX [1].
>
>         We have updated **Table 1** in the revised manuscript to include a column explicitly showing the latent channel size of our approach and competing baselines. Additionally, we have added baselines of our model with latent channel sizes of 8 and 16 for comparison. Please refer to **Table 1** in the updated manuscript.
>
>     - **We focused on a latent channel size of $4$** mainly for *fair comparisons with existing approaches, such as Open-Sora and Open-Sora-Plan, and to align with generative frameworks that use the same size*. Additionally, we fixed the latent channel size *to better highlight the contributions of our proposed modules across different spatio-temporal compression rates*.
>
> 2. **Why does our $8 \times 8 \times 8$ model give a superior performance compared to VideoGPT [2] ($4 \times 4 \times 4$)?**
>
>     - We would like to highlight that *VideoGPT uses discrete quantization, while our approach employs continuous Gaussian sampling*. As shown in previous works, such as LDM [3], continuous tokenization better preserves the semantic details of the input visual data compared to discrete tokenization with a predefined codebook, resulting in improved autoencoding performance. For instance, the experiments in Table 8 (page 21) of the LDM [3] paper demonstrate that the continuous model consistently outperforms the discrete one at the same or lower spatio-temporal compression rates. *Our results are consistent with this observation*.
>
>     - It is also important to note that the encoder in VideoGPT employs a 3D convolutional layer with a kernel size of $4 \times 4 \times 4$ to *directly* downsample the input video to the target latent size. In contrast, our approach utilizes *hierarchical* downsampling through the proposed FILM encoder and spatio-temporal downsampling modules, facilitating more effective encoding. This likely contributed to the superior performance of our approach, even with $8 \times$ more overall compression.
>
> 3. **Why is the LPIPS score of our $16 \times 8 \times 8$ slightly better than the LPIPS score of VideoGPT [2] ($4 \times 4 \times 4$)?**
>
>     - As noted in [2], the training setup of VideoGPT *does not include a perceptual (LPIPS) loss*, which likely contributes to its subpar performance on the LPIPS metric. In contrast, our training setup incorporates a perceptual loss, which helps explain the slightly better performance of our $16 \times 8 \times 8$ model on the LPIPS metric.
>     - *It is also worth noting that other baselines with higher compression rates, such as Open-Sora ($4 \times 8 \times 8$) and Open-Sora-Plan ($4 \times 8 \times 8$), which incorporate perceptual training, notably outperform VideoGPT ($4 \times 4 \times 4$) on the LPIPS metric, as shown in Table 1.*
>
> **References**
>
> [1] CogVideoX: Text-to-Video Diffusion Models with An Expert Transformer
>
> [2] VideoGPT: Video Generation using VQ-VAE and Transformers,
>
> [3] High-Resolution Image Synthesis with Latent Diffusion Models

---

> > ### Comment · Reviewer_MHdT · 2024-11-25
> > **Rebuttal Response**
> >
> > Since the authors have solved my problems, I'm glad to increase my rating to 6.

---

> > > ### Author Response · Authors · 2024-11-25
> > >
> > > Dear Reviewer MHdT,
> > >
> > > We are pleased to hear that our rebuttal addressed your concerns, and we sincerely thank you for raising your score.

---

### Official Review · Reviewer_tD7c · 2024-11-03

**Soundness:** 2
**Presentation:** 2
**Contribution:** 2
**Rating:** 5
**Confidence:** 4

**Summary:**

This paper introduces a video VAE for video compression and generation tasks. It features 4 major components: a causal 3D residual block, spatio-temporal downsampling module, spatio-temporal attention module, and FILM encoder. The proposed VAE is tested with the autoencoding and generation tasks.

**Strengths:**

The proposed method is evaluated with both the video/image autoencoding task and the video generation task. Moreover, extensive ablation studies were conducted to validate the effectiveness of the proposed method.

**Weaknesses:**

(1) The title “joint image and video compression” is misleading. I believe the task is focused primarily on video generation, not video transmission or compression (i.e. the autoencoding task). If the task is video compression/autoencoding, the resulting rate-distortion performance of the proposed method should be compared with that of the prior works on end-to-end learned video compression. However, this was not done when the authors reported results on the video/image autoencoding task. Just because the resulting latents have smaller dimensionality does not suggest that they would have smaller bit rates.

(2) In terms of the newly proposed components, I have the impression that their novelty is not very high.

(3) The necessity of causality is unclear in the context of video generation. If the aim is to enable both image and video generation, the image generation task is not explored in the current writing.

**Questions:**

(1) The title “joint image and video compression” is misleading. I believe the task is focused primarily on video generation, not video transmission or compression (i.e. the autoencoding task). If the task is video compression/autoencoding, the resulting rate-distortion performance of the proposed method should be compared with that of the prior works on end-to-end learned video compression. However, this was not done when the authors reported results on the video/image autoencoding task. Just because the resulting latents have smaller dimensionality does not suggest that they would have smaller bit rates.

(2) In Table 1, Video LDM (1 x 8 x 8) actually performs quite well particularly on DAVIS with large motion, as opposed to your better performing variant (4 x 8 x 8). Notably, Video LDM has relatively better TS performance. This leads to an impression that the proposed method cannot model well fast-motion sequences.

(3) The model size and MAC/pixel are currently missing in Table 1. Also, except TS, all the quality metrics are for images, not video. Why not use VMAF?

(4) I wonder if the competing methods adopt the GAN loss for training. This needs to be clarified in Table I. In addition, it is unclear how much the GAN loss contributes to the performance of the proposed method.

(5) For the image autoencoding task, how is the proposed method used?

(6) For the video generation task, it is unclear whether the proposed VAE is trained or fine-tuned together with the diffusion model. It appears to me that the proposed VAE is pre-trained to generate latents that follow a simple Gaussian distribution.

(7) In addition, the proposed VAE can be a stand-alone approach to video generation. I wonder how it performs as compared to the diffusion-based modeling of the video latents.

(8) In Figure 4, I wonder whether the proposed method can generate well fast-motion sequences.

(9) FILM encoder is shown to be effective in the ablation study. But, the proposed VAE does not appear to work well on the autoencoding task in Table I, as compared to Video LDM.  This is a bit confusing.

**Details Of Ethics Concerns:**

Not applicable.

---

> ### Author Response · Authors · 2024-11-21
> **Response to Reviewer tD7c [1/2]**
>
> Thank you for your valuable review and constructive suggestions. We have carefully addressed your concerns and revised our paper accordingly, as outlined below.
>
> ## Response to "Weaknesses"
>
> - **(1)  Title could be misleading:**
>
>     We appreciate the reviewer’s concern.  We would like to clarify that our paper consistently specifies that the term “*compression*” is used in the context of *generation*. Additionally, in **Related Works** section (refer to **Appendix B**), *we provide an in-depth discussion on both general video compression and video compression for generation*. It is also worth noting that “*video compression*” is frequently used in recent literature as a synonym for “*video autoencoding*” [1,2,3,4]. **To prevent any potential misunderstandings, we updated our paper’s title, replacing “compression” with “autoencoding”**. Please check the revised manuscript.
>
> - **(2) Regarding novelty:**
>
>     As acknowledged by the other reviewers, we would like to emphasize that the primary contribution of our work is identifying three key issues in existing video VAEs, as outlined in the **Introduction** section (**Lines 63–96**), and proposing an effective method to address them. The effectiveness of our approach has been validated through extensive ablation studies, as shown in **Table 4**.
>
> - **(3) Necessity of causality:**
>
>     Although our work primarily focuses on video autoencoding for generative modeling, the causality formulation is inspired by recent related works [1,2,3,4] and it enables our approach to be flexibly applied to both image and video autoencoding, as demonstrated in Table 1 and Table 2. **The ability to jointly encode images and videos has been shown in previous works [1,2] to be crucial for improving video generation quality**. Additionally, it enables other applications such as image-to-video generation and variable-length video generation [1,2,3].
>
> ## Response to "Questions"
>
> - **(1) About title of the paper:**
>
>    Please refer to our response to **question (1)** under **Weaknesses**
>
> - **(2) Comparison with VideoLDM on large motions:**
>
>     We would like to clarify that VideoLDM ($1 \times 8 \times 8$) employs a temporal reasoning layer to align latents from individual frames but **does not perform any temporal compression**. Consequently, it is expected to perform well on the temporal smoothness (TS) metric, particularly for large-motion (low frame-rate) datasets such as DAVIS. In comparison, our model ($4 \times 8 \times 8$) achieves $4 \times $ greater temporal compression while still achieving a strong performance on the TS metric.
>
>     **For a fairer comparison, we have included the results of our $1 \times 8 \times 8$ model in Table 1** in the revised manuscript. *As can be inferred from the table, our ($1 \times 8 \times 8$) model significantly outperforms VideoLDM ($1 \times 8 \times 8$) on temporal smoothness (TS) metric across both DAVID and Xiph-2K datasets*.
>
> - **(3) Computational complexity and video-based metrics:**
>
>
>     We have included a comparison of computational complexity, including model size and inference speed, between the proposed model and competing approaches in **Table 7** in the appendix. A related discussion has also been added to **Appendix A.5**.
>
>
>     We have also included two additional **video-based** metrics in **Table 1** in the updated manuscript: *reconstruction FVD (rFVD)* and *STREAM [5]*, which evaluate the spatio-temporal perceptual quality and fidelity of the decoded videos. As shown in Table 1, *our approach consistently outperforms competing methods on the newly added video-based metrics*, aligning with the superior performance observed using other metrics.
>
>     **P.S.** We attempted to include the VMAF metric in our comparisons but could not find an open-source PyTorch-based implementation. The official codebase [6] posed challenges due to its restrictive requirements, such as specific video formats and resolutions. Consequently, we opted for more commonly used and easily reproducible metrics.
>
> - **(4) Regarding GAN loss:**
>
>     We would like to point out that GAN loss is a standard loss in training image/video autoencoders and *all the competing approaches listed in Table 1 are trained with GAN loss included*.
>
>     To show the contribution of GAN loss in our proposed model, we report the performance of our proposed method with and without GAN training in **Table 9** in the appendix. Our experimental analysis shows that GAN training primarily enhances the perceptual quality of the decoded videos, as evidenced by a slight degradation in metrics like LPIPS and rFVD when the model is trained without GAN loss.
>
>     *This discussion has been added to **Appendix A.8** of the revised manuscript.*

---

> > ### Author Response · Authors · 2024-11-21
> > **Response to Reviewer tD7c [2/2]**
> >
> > - **(5) Image autoencoding:**
> >
> >     For image autoencoding, we simply pad the image in the temporal dimension and feed it to our model. We only consider the first reconstructed frame as an output and ignore the rest of the frames. The padding type did not matter since our model is temporally causal.
> >
> > - **(6) Regarding the video generation task:**
> >
> >     All autoencoders, including ours and the baselines, are kept *frozen* during the generation experiments. We have clarified this in the revised manuscript in **Lines 395-396**.
> >
> > - **(7) About proposed VAE :**
> >
> >     We are a bit unclear about this comment. Could the reviewer please clarify what he/she meant by “**the proposed VAE can be a stand-alone approach to video generation**”?
> >
> >     Our model is a continuous autoencoder, and the latents it encodes (after being trained) can serve as inputs for training diffusion-based video generation models more efficiently, due to their significantly reduced dimensionality compared to the original video resolutions.
> >
> > - **(8) Generating fast-motion sequences:**
> >
> >     As shown in Table 1, our model demonstrates strong autoencoding performance on fast-motion datasets such as DAVIS and Xiph-2K, outperforming competing methods. This suggests that our model's latents are sufficiently expressive for accurate reconstruction of input videos with large motions. *Thus, it is reasonable to expect that generative training on large-motion videos using our model as an autoencoder would enable the generation of fast-motion videos.*
> >
> >     For example, on the challenging UCF-101 dataset, which contains numerous human activity videos with fast motions, our autoencoder, paired with the generative model from Open-Sora, achieves strong performance compared to competing approaches, as shown in Table 3. We also encourage reviewing the generated video samples provided in the **supplemental video**.
> >
> > - **(9) Comparison with VideoLDM:**
> >
> >     Please refer to our answer to **question (2)** under **Questions**.
> >
> > **References**
> >
> > [1] Phenaki: Variable Length Video Generation from Open Domain Textual Descriptions
> >
> > [2] W.A.L.T: Photorealistic Video Generation with Diffusion Models
> >
> > [3] Sora: Video generation models as world simulators
> >
> > [4] Magvit-v2: Language Model Beats Diffusion -- Tokenizer is Key to Visual Generation
> >
> > [5] STREAM: Spatio-TempoRal Evaluation and Analysis Metric for Video Generative Models
> >
> > [6] https://github.com/Netflix/vmaf

---

> > > ### Author Response · Authors · 2024-11-26
> > > **Looking Forward to Your Feedback**
> > >
> > > Dear Reviewer tD7c,
> > >
> > > Thank you once again for your constructive review and valuable suggestions. We have carefully addressed your concerns and revised our manuscript accordingly, as detailed in our response above. We are writing to confirm if there are any remaining issues we can address **before the discussion period concludes**. We look forward to your reply.

---

> ### Comment · Reviewer_tD7c · 2024-11-28
> **More questions**
>
> I thank the authors for taking much effort in addressing my comments. Most of them have been addressed properly. Some further questions are as follows:
>
> (2)  Comparison with VideoLDM on large motions
> It appears that temporal compression has a significant impact on generated video quality particularly for videos with large motion. This leads me to the question whether temporal compression is impractical for real-world applications, especially when one like to generate videos with complex motion. Under what circumstances should this temporal compression be used? What are the benefits of performing temporal compression?
>
> (3) Computational complexity and video-based metrics
> In Table 7 of the appendix, how come the two variants of the proposed method with different Ch. sizes have the same model size.
>
> (7) About proposed VAE
> I mean that at inference time, you may as well draw a Gaussian sample and do the video generation. In this case, you do not train another diffusion model to learn the prior distribution for the pre-trained VAE. I am curious how the video quality would be if you used the Gaussian sample to generate videos.

---

> ### Author Response · Authors · 2024-11-28
> **Response to Follow-up Questions**
>
> Dear Reviewer tD7c,
>
> Thank you for your response. We are pleased to hear that most of your comments have been addressed. Below, we carefully address the follow-up questions you raised.
>
> - **(2) Regarding temporal compression:**
>
>     We agree with the reviewer that temporal compression can affect the quality of the decoded video, *particularly when the video has a low frame rate or involves highly complex motion*. This reflects the general trade-off between spatio-temporal compression and the quality of the decoded video in generative modeling.
>
>     However, most user-generated videos in internet traffic—commonly used to train video generation models—do not exhibit highly complex motions, tend to have relatively high frame rates, and contain significant temporal redundancy. For example, the recently proposed OpenVid-1M dataset [1] contains videos with frame rates ranging from 30-60 fps. Additionally, a recent study [2] shows that most video datasets used for generative training exhibit relatively small dynamics. In such cases, *temporal compression can still be effectively leveraged.*
>
>     More importantly, *temporal compression improves the computational efficiency of long video generation during both training and inference*. For instance, when latents are encoded with a $1 \times 8 \times 8$ model, each generated latent corresponds to a single frame. However, encoding with an $8 \times 8 \times 8$ model results in decoding 8 frames from each generated latent, allowing for the generation of much longer videos with the same computational cost.
>
> - **(3) Regarding the model size of the two variants:**
>
>     We would like to clarify that the two model variants *do not have exactly the same size*. The model with a latent channel size of 4 has a total of 243,384,571 parameters, while the model with a latent channel size of 16 has a total of 243,385,555 parameters. *The sizes listed in Table 7 of the appendix appear identical due to rounding to the nearest $N \times 10^6$, where N=243.4*. We will clarify this in the final version of our paper.
>
>     We would like to emphasize that the encoder-decoder architecture of both models is identical. The only difference is *the output channel size of the Causal3DConv layer* before Gaussian sampling and *the input channel size of the Causal3DConv layer* after Gaussian sampling in the Middle Block. Please refer to Figure 1 for further details.
>
> - **(7) Regarding the proposed VAE:**
>
>     Thank you for providing a clarification to the question.
>
>     In response, we would like to highlight a subtle distinction between *VAEs used for direct generation [3, 4]* and *those designed to create a meaningful latent space for other generative models [5, 6]*. Specifically, VAEs for compression often employ a smaller KL divergence term.
>
>     In a Variational Autoencoder (VAE), the loss function consists of two key components: the reconstruction loss, which ensures accurate reconstruction of the input, and the KL divergence loss, which regularizes the latent space to align with a standard Gaussian distribution. In line with the approach described in the seminal work LDM [5], we train our video VAE with a small KL divergence coefficient of $1e-6$. This low weight significantly reduces the impact of the KL divergence term, causing the latent space to deviate notably from a standard Gaussian distribution. *As a result, sampling from Gaussian distribution is unable to generate meaningful videos as the VAE is dominated more by the reconstruction term and behaves more like an autoencoder*.
>
>
> **References**
>
> [1] OpenVid-1M: A Large-Scale High-Quality Dataset for Text-to-video Generation
>
> [2] Evaluation of Text-to-Video Generation Models: A Dynamics Perspective
>
> [3] NVAE: A Deep Hierarchical Variational Autoencoder
>
> [4] An Introduction to Variational Autoencoders
>
> [5] High-Resolution Image Synthesis with Latent Diffusion Models
>
> [6] Sora: Video generation models as world simulators

---

> > ### Author Response · Authors · 2024-11-30
> > **Follow-Up: Looking Forward to Your Feedback**
> >
> > Dear Reviewer tD7c,
> >
> > Thank you once again for your thoughtful response to our rebuttal and for raising your score. As the **author-reviewer discussion period nears its conclusion**, we would appreciate your confirmation on whether your follow-up questions have been fully addressed or if you have any additional questions. We look forward to your feedback.
> >
> > Paper 2418 Authors

---

### Official Review · Reviewer_LuUv · 2024-11-03

**Soundness:** 3
**Presentation:** 3
**Contribution:** 4
**Rating:** 8
**Confidence:** 4

**Summary:**

The paper introduces a novel architecture for 3D Causal VAE aimed at image and video compression, addressing three key issues found in existing video VAE architectures:

1. Handling Small and Fast-Moving Objects at Higher Compression Rates: Standard encoders struggle with these objects. The authors tackle this by proposing a FILM-based encoder.

2. Downsampling and Upsampling Challenges: Non-learnable downsampling loses high-frequency spatiotemporal (ST) information, while learnable downsampling tends to overfit. To address this, the authors propose a novel Down and Upsampling layer design that employs a dual-path network with parallel branches utilizing both learnable and non-learnable kernels.

3. Motion Reconstruction Quality and Smoothness: To enhance these aspects, the authors introduce a novel Flow Regularization loss that facilitates in Video VAE training.

The authors trained their 3D VAEs at various compression rates (T × H × W) such as 4 × 8 × 8 (256), 8 × 8 × 8 (512), 16 × 8 × 8 (1024), and 16 × 16 × 16 (4096). Their experiments showed strong results when compared to other baselines.

Overall, this is a good paper that introduces novel architectural decisions for Video-VAE and shows strong experimental results. However, it would benefit from including additional experimental results and details (See Weaknesses and Questions).

**Strengths:**

1. Importance: This paper addresses a critical issue of training a high-quality VAE for image and video compression.

2. Novelty: The paper introduces a unique and previously unexplored approach to tackle the aforementioned problem.

3. Results Quality: Experimental results demonstrate strong performance compared to existing baselines. Notably, the paper provides results for very high video compression rates (16 × 8 × 8 and 16 × 16 × 16).

4. Clarity: The main text of the paper is well written and easy to follow.

5. Ablations: Ablation studies clearly illustrate the positive impact of the proposed architectural decisions.

**Weaknesses:**

1. Missed comparisons: Tables 1 and 2 do not include comparisons with CogVideoX VAE [1] and CV-VAE [2], which are significant baselines in this field.

2. Video quality metrics: Table 1 reports only frame-wise video quality metrics and neglects any temporal consistency metrics. It is recommended to measure the Fréchet Video Distance (FVD) between original and reconstructed videos (rFVD) to provide a more comprehensive evaluation.

3. Video generation results: The video generation results in Table 3 were conducted on relatively simple video datasets such as SkyTimelapse and UCF-101. It is recommended to include experiments on more complex datasets and tasks, such as text-conditional video generation on the WebVid dataset, to better assess the proposed method's capabilities.

4. Evaluation resolution: The authors have not specified the evaluation resolution for videos in Tables 1 and images in Table 2, which is crucial for assessing the results accurately.

5. Analysis on different resolutions: The paper lacks an analysis of how the proposed VAE performs across different resolutions compared to other baselines. The performance on high-resolution videos (e.g., 720p) is particularly relevant and should be explored.

6. Inference time: Information about the inference time of the models in Tables 1 and 2 is missing. Including this data would provide a better understanding of the practical utility of the proposed method.

7. Method’s limitations: The paper does not provide an analysis of the limitations of the proposed method, which is necessary for a comprehensive evaluation.

**Questions:**

My main concerns and comments regarding the paper are related to the experiment section:

1. Tables 1 and 2 would benefit from including results for CogVideoX VAE [1] and CV-VAE [2], adding rFVD metric,  providing information about evaluation resolution and inference time of the models. Please refer to weaknesses 1, 2, 4, and 6 for more details.

2. The paper would benefit from adding limitation analysis of the proposes method. Especially interesting to see how the method generalizes to different video resolutions (128x128, 256x256, 512x512, 480p, and 720p) in comparison with other methods. For more details see weaknesses 5 and 7.

3. Table 3 provides video generation results for relatively simple datasets and tasks. It is recommended to test the model on text-conditional video generation on the WebVid dataset if possible. Refer to weaknesses 3 for more details.

[1] Yang, Z., Teng, J., Zheng, W., Ding, M., Huang, S., Xu, J., ... & Tang, J. (2024). CogVideoX: Text-to-video diffusion models with an expert transformer. arXiv preprint arXiv:2408.06072.

[2] Zhao, S., Zhang, Y., Cun, X., Yang, S., Niu, M., Li, X., ... & Shan, Y. (2024). CV-VAE: A Compatible Video VAE for Latent Generative Video Models. arXiv preprint arXiv:2405.20279.

---

> ### Author Response · Authors · 2024-11-21
> **Response to Reviewer LuUv**
>
> Thank you for your insightful review and valuable feedback. We have thoroughly considered your suggestions and made the necessary revisions to our paper, as outlined below.
>
> - **1. Missed comparisons:**
>
>     Thank you for the suggestion. We have updated our manuscript by including both CogVideoX VAE and CV-VAE in our experimental comparisons. Please refer to **Table 1** and **Table 2** in the revised manuscript.
>
> - **2. Video quality metrics:**
>
>     We have also added rFVD results in **Table 1**. Please refer to the revised manuscript.
>
> - **3. Video generation results:**
>
>     We would like to highlight that utilizing SkyTimelapse and UCF101 for generation experiments is a standard practice commonly followed in related works.
>
>     We appreciate the reviewer’s suggestion to conduct large-scale text-to-video generation experiments. However, these experiments are beyond the scope of our work due to **current resource constraints** and are left for future exploration. *To support such efforts, we will open-source our code and pretrained checkpoints*.
>
> - **4. Evaluation resolution:**
>
>     The resolution used in Table 1 is $480$p. For Table 2, we used $256 \times 256$ resolution. We have added this clarification in the updated manuscript (**Lines 305-307**).
>
> - **5. Analysis on different resolutions:**
>
>     Following the reviewer's suggestion, we have conducted a performance analysis of our approach and competing baselines across various video resolutions: $128 \times 128$, $256 \times 256$, $480$p, and $720$p. The results are presented in **Table 7** in the appendix, with a related discussion included in **Appendix A.4**.
>
> - **6. Inference time:**
>
>    We have included the inference time comparison of our approach and baseline models in **Table 7** in the appendix, along with a related discussion in **Appendix A.5**.
>
> - **7. Method’s limitations:**
>
>     While our work advances the state-of-the-art in video autoencoding, particularly at higher spatio-temporal compression rates, it does have a few limitations. Our experimental analysis reveals that temporal jittering can occur between decoded frames at very high compression rates, such as $16 \times 16 \times 16$. Qualitative examples of such cases are provided in the supplementary video. While this phenomenon is expected due to the inherent challenges of extreme compression, we argue that our low-resolution training also contributes to this limitation.
>
>     For an input clip of size $17 \times 128 \times 128 \times 3$ and a given compression rate of $16 \times 16 \times 16$, the encoded latent will have a dimension of $2 \times 8 \times 8 \times 4$. Due to its significantly smaller size, the encoded latent may not capture all the information needed for the decoder to reconstruct the input video. While the straightforward solution is to use higher resolution clips during training, this was computationally infeasible for our approach due to the significant cost of the quadratic scale attention in the spatio-temporal attention blocks (*STAttnBlock*). For instance, using a $256 \times 256$ resolution increases the token count by a factor of 4. Removing the attention blocks from our model is not a viable solution either, as our ablation experiments in Table 4(d) have shown them to be a crucial component of the proposed video VAE.
>
>     Therefore, a promising direction for future work to address this issue would be to explore more computationally efficient spatio-temporal attention mechanisms, such as Mamba [1], for higher-resolution training. This could, in turn, enable more robust higher spatio-temporal compression while mitigating the associated computational challenges.
>
>     *This discussion has been added to **Section 5** of the revised manuscript.*
>
>
> **References**
>
> [1]  Mamba: Linear-Time Sequence Modeling with Selective State Spaces

---

> > ### Comment · Reviewer_LuUv · 2024-11-25
> >
> > Dear Authors,
> >
> > Thank you for addressing my major concerns and including additional information in the manuscript's revision. I believe this paper makes a valuable contribution to the research field, and I am pleased to raise my score accordingly.
> >
> > Best regards,
> >
> > Reviewer LuUv

---

> > > ### Author Response · Authors · 2024-11-28
> > >
> > > Dear Reviewer LuUv,
> > >
> > > Thank you for your response and for raising the score. We are pleased to hear that our rebuttal addressed your concerns and appreciate your acknowledgment of the valuable contribution our paper makes to the research field.

---

### Official Review · Reviewer_76n7 · 2024-11-04

**Soundness:** 3
**Presentation:** 3
**Contribution:** 3
**Rating:** 8
**Confidence:** 5

**Summary:**

The paper presents a novel approach to joint image and video compression via a causal variational autoencoder (VAE) that leverages 3D convolutional and self-attention mechanisms. Designed to tackle the challenge of temporal and spatial redundancy in video data, the proposed model incorporates a dual-path architecture for downsampling and upsampling, as well as a flow regularization loss to maintain temporal coherence. Key contributions include a scale-agnostic encoder, a dual-path network for robust spatio-temporal feature extraction, and a loss function to enhance motion preservation in compressed sequences. The experimental results show the model’s superiority in video fidelity and compression rates over existing methods, particularly in handling large motions and maintaining temporal consistency.

**Strengths:**

Hybrid Approach: The paper successfully integrates causal 3D convolutions with self-attention layers to handle temporal and spatial redundancy in video compression, creating a versatile model capable of high-quality compression for both images and videos. This hybrid approach is particularly relevant as it extends beyond existing methods that often separate these tasks or overlook temporal redundancy.
Robust Encoder-Decoder Design: By leveraging a dual-path downsampling and upsampling mechanism, the proposed model achieves a high compression rate without significant quality loss, evidenced by its superior PSNR and SSIM scores in Table 1. Additionally, the FILM encoder’s ability to handle large motion in compressed videos aligns well with contemporary needs in video compression.
Comprehensive Evaluation: The paper rigorously evaluates its model across multiple datasets and compression settings, supporting its claims with ablation studies and comparisons to several state-of-the-art methods. The qualitative results further emphasize its performance in preserving motion fidelity and detail, especially for fast-moving objects.

**Weaknesses:**

* The paper should cite the recent ECCV 2024 work, Hybrid Video Diffusion Models with 2D Triplane and 3D Wavelet Representation, as this paper also explores autoencoder backbones and addresses related challenges in video compression. A comparative analysis, including quantitative and qualitative evaluations against this baseline, would strengthen the evidence for the proposed model’s advantage.
* The claim that the dual-path network is "temporally agnostic and can encode and decode arbitrarily long videos at varying lengths" may be overstated. The performance improvements in Table 4’s ablation study do support better temporal adaptability; however, the reliance on learnable kernels suggests some degree of generalization error remains, as evidenced by performance drops with different sequence lengths. It would be more accurate to moderate this claim, acknowledging the limitations indicated by remaining performance variability.
* Given the paper’s focus on maintaining high spatio-temporal quality in compressed videos, metrics specifically targeting spatio-temporal fidelity must be also compared, such as STREAM (ICLR 2024 STREAM: Spatio-TempoRal Evaluation and Analysis Metric for Video Generative Models). These metrics could provide a more nuanced understanding of the model’s strengths and areas needing improvement in terms of temporal coherence and spatial detail.
* The model’s large size and complexity might raise concerns about overfitting, particularly when evaluated on datasets like UCF101 and SkyTimelapse. To address this, a thorough analysis for potential overfitting or memorization is recommended, especially in cases where generated samples closely resemble training data. Expanding the qualitative analysis in Figure 4 to include more diverse sample comparisons would help clarify this aspect.
* Table 1 demonstrates the model’s robustness at more severe compression rates (e.g., 4\times8\times8 and 8\times8\times8), but the performance at commonly used settings such as 1\times8\times8 and 4\times4\times4 is not discussed. Adding a direct comparison at these standard settings would provide a more balanced view, allowing for an "apple-to-apple" comparison with other models. Such an addition could reveal how the model’s advantages scale across different compression intensities, offering deeper insights into the model’s comparative performance under varying constraints.
* To provide readers with an estimation of computational resources and convergence speed, including additional details on training time until convergence would be beneficial. This information would help contextualize the model’s practical application feasibility.
* Please elaborate more on 3D Model Extension in GAN Training.
* Although the introduction mentions experiments on noise corruption and varying sequence lengths, these are not included in the main text. Readers may find it disorienting without a clear reference directing them to the Appendix for these analyses. Please note this clearly.
* Is normalization applied during the concatenation of FILM’s outputs? Wouldn't it be problematic if they are not normalized (due to different output statistics from different scales)?
* Any specific reason for using 1+T over T?
* In the conclusion, "3 key contributions" -> "three key contributions"

**Questions:**

Please see Weaknesses

---

> ### Author Response · Authors · 2024-11-21
> **Response to Reviewer 76n7 [1/2]**
>
> Thank you for your thoughtful review and constructive comments. We have carefully addressed your suggestions and revised our paper accordingly, as detailed below.
>
> - **Cite and compare with HVDM (ECCV'24):**
>
>     Thank you for bringing HVDM (ECCV'24) [1] to our attention. We have updated our manuscript to include a citation and incorporate it into our experimental comparisons (**Table 1**). *Since pretrained checkpoints for HVDM were not made publicly available, we trained their model from scratch adhering to the training code provided in their official repository*. As highlighted in **Table 1** of the updated manuscript, our $1 \times 8 \times 8$ and $4 \times 8 \times 8$ models significantly outperform HVDM ($1 \times 8 \times 8$), while our $8 \times 8 \times 8$ model demonstrates highly competitive performance. Additionally, qualitative comparisons of our approach with HVDM have been added to the **Figure 8** in the appendix.
>
> - **Clarifications on the usage of "temporally agnostic'':**
>
>      Following the reviewer’s suggestion, we have toned down the claim and replaced “*temporally agnostic*” with “*better temporal adaptability*” throughout our paper. Please refer to **Lines 87-89**, **Lines 188-189**, and **Lines 477-479** in the revised manuscript.
>
> - **Evaluation using STREAM metrics:**
>
>     Thank you for the suggestion. We have included the metrics from STREAM [2] in our experimental evaluation. Specifically, we report the **average** of the spatio-temporal fidelity (**STREAM-F**) and temporal flow (**STREAM-T**) scores for our models and competing approaches. The updated results are presented in **Table 1** of the revised manuscript. As shown in the table, *our approach achieves superior spatio-temporal fidelity and temporal flow compared to competing methods*, aligning with the performance trends observed using other metrics in Table 1.
>
> - **Possibility of overfitting on UCF101 and SkyTimelapse:**
>
>     We appreciate the reviewer’s concern and clarify that *our pretrained video VAE remains frozen during the generation experiments, ensuring it cannot overfit to SkyTimelapse or UCF-101, as these datasets were not part of its training*. However, we acknowledge the reviewer’s point that training with Open-Sora’s generative framework may still be susceptible to overfitting.
>
>     To ensure a fair comparison, we have demonstrated in Table 3 that our video VAE significantly outperforms Open-Sora’s own VAE in video generation experiments. Additionally, to address this concern further, we have included diverse video samples generated by Open-Sora using both our video VAE and Open-Sora’s own VAE in the **supplemental video**. We have also added more generated samples in **Figure 4**.
>
> - **Adding baselines for an apples-to-apples comparison:**
>
>     We have added the performance of our $1 \times 8 \times 8$ and $4 \times 4 \times 4$ models in **Table 1** of the revised manuscript. As anticipated, reducing spatio-temporal compression significantly improves autoencoding performance.
>
> - **Discussion on computation resources and convergence speed:**
>
>     Our experiments identified three key factors affecting computational resource usage and convergence speed: *spatio-temporal attention blocks*, *training video resolution*, and *batch size*. Our default configuration uses a video resolution of $17 \times 128 \times 128 \times 3$, spatio-temporal attention blocks in both the encoder and decoder networks, and a batch size of 48. On 48 A100 (40 GB) GPUs, this setup converges in about 96 hours. Removing spatio-temporal attention layers reduces convergence time to 40 hours, but as shown in the ablation study (Table 4d), it significantly degrades performance, underscoring the trade-off between efficiency and model quality. The quadratic scaling of attention in these blocks is the main factor driving the longer training times. Increasing resolution to $17 \times 256 \times 256 \times 3$ was infeasible on 40 GB GPUs due to the $4 \times$ increase in token count, which greatly increases the computational demands of spatio-temporal attention. Smaller batch sizes, such as 8 on 8 A100 GPUs, extended training time to 144 hours to match the performance of the default setting.
>
>    *We have included this discussion in the new **Appendix A.6** and provided a clear reference to it in the main paper (**Line 293**)*
>
> - **Elaborate more on the GAN architecture:**
>
>     We have expanded on the details of the 3D convolution-based discriminator used in our work. Please refer to **Lines 278-281** in the updated manuscript for further information.

---

> ### Author Response · Authors · 2024-11-21
> **Response to Reviewer 76n7 [2/2]**
>
> - **Missing references directing to the appendix:**
>
>     Thanks for pointing this out. We have added clear references to the appendix in the revised manuscript. Please refer to **Lines 124-128**
>
> - **Regarding concatenation of FILM’s outputs:**
>
>     We experimented with different methods of combining FILM's output, including *normalizing before concatenation* and *passing each output through a 3D Conv layer*. However, we did not observe any significant performance improvement with these approaches. Therefore, *we chose direct concatenation for simplicity*. Our intuition is that any potential statistical differences in the concatenated latent are addressed during training through the *Middle Block* layers before Gaussian sampling (refer to Figure 1).
>
> - **Reason for using $1+T$ over $T$:**
>
>     The main reason for using $1+T$ instead of $T$ is to maintain a temporally causal formulation throughout the encoder and decoder, particularly during downsampling and upsampling, *enabling the joint encoding of images and videos within a single model*. This approach is commonly adopted in recent works [3, 4].
>
> - **"3 key contributions" -> "three key contributions":**
>
>     Thank you for pointing this out. We have corrected it in the revised manuscript. Please refer to **Line 533**.
>
>
> **References**
>
> [1] HVDM: Hybrid Video Diffusion Models with 2D Triplane and 3D Wavelet Representation
>
> [2] STREAM: Spatio-TempoRal Evaluation and Analysis Metric for Video Generative Models
>
> [3] Phenaki: Variable Length Video Generation from Open Domain Textual Descriptions
>
> [4] Magvit-v2: Language Model Beats Diffusion -- Tokenizer is Key to Visual Generation

---

> > ### Comment · Reviewer_76n7 · 2024-11-24
> >
> > Dear Authors,
> >
> > Thank you for the detailed responses. I am satisfied with the rebuttal and raise my score. I think the practical value of the paper is good enough and well supported by the experiments. Still, it is very unfortunate that it would be quite hard for many of the academic groups to reproduce or compare the results due to very high computational requirements for training. Do you think the current framework can become scalable with a more smaller model size?
> >
> > Best,
> > Reviewer 76n7

---

> > > ### Author Response · Authors · 2024-11-25
> > >
> > > Dear Reviewer 76n7,
> > >
> > > Thank you for your prompt reply and for raising the score. We are pleased to hear that you found our rebuttal satisfactory and appreciate your recognition of the practical value of our work. Below, we address the raised question.
> > >
> > > - *Do you think the current framework can become scalable with a more smaller model size?*:
> > >
> > >     We firmly believe that the core contributions of our work—the FILM encoder, temporally adaptable spatio-temporal down/upsampling blocks, and flow regularization—are not closely tied to model size and should therefore enhance the performance of smaller video VAEs as well. As discussed in **Section 5** and **Appendix A.6** of the revised manuscript, the primary computational bottleneck in our model lies in the spatio-temporal attention blocks, which also play a notable role in enhancing network performance, as demonstrated in Table 4d. As mentioned in **Section 5**, a promising direction for future work would be to explore more computationally efficient spatio-temporal attention mechanisms, such as Mamba [1], for video VAE architectures. This could alleviate current computational challenges and enhance the scalability of our framework. We will elaborate on this further in the final version of our paper.
> > >
> > > **References**
> > >
> > > [1] Mamba: Linear-Time Sequence Modeling with Selective State Spaces

---

### Author Response · Authors · 2024-11-24
**Follow-Up on Author-Reviewer Discussion**

Dear Reviewers,

We sincerely thank you once again for the time and effort you have dedicated to reviewing this paper. Your invaluable feedback have significantly contributed to improving its quality.

In the revised version, we have polished the manuscript, incorporated additional experimental results, and addressed your concerns with detailed clarifications. **As the deadline for the Author-Reviewer discussion is approaching**, we would like to ensure that our responses have sufficiently addressed your feedback. If there are any remaining concerns or additional clarifications or experiments that you would like us to provide, please do not hesitate to let us know.

Thank you once again for your time and thoughtful input.

Best regards,

Paper 2418 Authors

---

> ### Comment · Area_Chair_GuWq · 2024-11-27
>
> Dear Reviewers,
>
> Thanks for your contributions in reviewing this paper.
>
> As the author-reviewer discussion deadline is approaching, please could you take a look at the authors' rebuttal and see if it addressed your concerns or if you have any further questions. Please feel free to start a discussion.
>
> Thanks,
>
> AC

---

### Meta-Review · Area_Chair_GuWq · 2024-12-20

**Metareview:**

In this paper, the authors presented a new joint image and video compression method. Specifically, based on a variational autoencoder (VAE), causal 3D convolution, spatio-temporal down/upsampling block, and a flow regularization loss were used to achieve the goal. Experimental evaluations show the effectiveness of the proposed method over the compared previous methods. The main strengths of this paper are:
- The studied problem is important for image and video compression.
- The proposed technical approach is novel with the FILM encoder that handle large motion.
- Extensive experimental analysis across multiple datasets and different compression settings showed the effectiveness of the proposed method and supported the claims made. Additional ablation studies provide further supporting evidence for the proposed method. Qualitative results highlighted the strong performance of the proposed method in preserving motion details.

The weaknesses of this paper include:
- Details of some technical designs and experimental settings are missing, making the paper a bit hard to follow in some cases and resulting in potential misleadings.
- Missing comparison and experiments to validate certain points, e.g. comparison to related works and spatio-temporal metrics, in-depth analysis for overfitting, training time etc.
- Further analysis on video generation, e.g. text-conditional video generation, is needed to better assess the capability of the proposed method
- One reviewer was concerned about the novelty and contributions of this paper.

Overall, this paper addressed an important problem in video compression, by presenting a new method with strong performance. The corresponding experimental evaluations and ablation study also provide sufficient evidence and insights in this field. There were some concerns raised by the reviewers but most of them are minor and has been addressed during the rebuttal phase. As a result, the AC is happy to recommend an Accept, but would encourage the authors to incorporate the additional clarifications and evidence into the final version of the paper.

**Additional Comments On Reviewer Discussion:**

This paper received review comments from five expert reviewers. During the rebuttal phase, the authors managed to provide detailed responses to each of the reviewers. There was a long back-and-forth discussion. Through the authors-reviewers discussions, four reviewers decided to increase their ratings, ending up with a generally positive rating of 2 Accept, 2 borderline Accept, and 1 borderline Reject. All reviewers agreed on the main strengths and weaknesses of this paper. Although there are some limitations, they are relatively minor, and considering the potential contributions this paper may bring to the community, a final decision of Accept was made according to the above points.

---

### Decision · Program_Chairs · 2025-01-22

Accept (Poster)